# Identification of the Fractional Zener Model Parameters for a Viscoelastic Material over a Wide Range of Frequencies and Temperatures

**DOI:** 10.3390/ma14227024

**Published:** 2021-11-19

**Authors:** Zdzisław M. Pawlak, Arkadiusz Denisiewicz

**Affiliations:** 1Institute of Structural Analysis, Poznan University of Technology, Ul. Piotrowo 5, 60-965 Poznan, Poland; zdzislaw.pawlak@put.poznan.pl; 2Division of Structural Mechanics, University of Zielona Góra, Ul. Licealna 9, 65-417 Zielona Góra, Poland

**Keywords:** viscoelastic materials, fractional derivatives, frequency and temperature dependence, identification of rheological parameters

## Abstract

The paper presents an analysis of the rheological properties of a selected viscoelastic material, which is dedicated to the reduction of vibrations in structures subjected to dynamic loads. A four-parameter, fractional Zener model was used to describe the dynamic behavior of the tested material. The model parameters were identified on the basis of laboratory tests performed at different temperatures and for different vibration frequencies. After proving that the material is thermoreologically simple, the so-called master curves were created using a horizontal shift factor. The Williams–Landel–Ferry formula was applied to create graphs of the master curves, the constants of which were determined for the selected temperature. The resulting storage and loss module functions spanned several decades in the frequency domain. The parameters of the fractional Zener model were identified by fitting the entire range of the master curves with the gradientless method (i.e., Particle Swarm Optimization), consisting in searching for the best-fitted solution in a set of feasible solutions. The parametric analysis of the obtained solutions allowed for the formulation of conclusions regarding the effectiveness of the applied rheological model.

## 1. Introduction

Viscoelastic (VE) materials are often used to reduce excessive vibration in building structures caused by earthquakes or wind action. They are most often used in the so-called passive damping systems [1,2,3,4,5,6], where they are selected to effectively dissipate energy during forced vibrations. The advantage of passive damping systems is that they are easy to implement and maintain, and there is no need to supply additional energy to the system during operation. The limitation of passive dampers is the relatively small frequency range in which they are effective. Therefore, passive dampers should be selected so that they reduce well the vibrations corresponding to the lowest natural frequencies of the considered structure. On the other hand, the dynamic parameters of most viscoelastic materials used in vibration dampers depend on temperature, frequency and amplitude of forcing vibrations. For this reason, the identification of rheological properties of a viscoelastic material taking into account the above influences is a challenge that is the subject of numerous studies, e.g., [7,8,9].

The dynamic behavior of a VE material or a passive damper as a device can be described by the equation of motion, which in turn can be related to the so-called mechanical model. In the literature, there are classical models made of properly connected dashpots and springs, as well as the so-called fractional models, in which there is an element with viscous and elastic properties described with the use of non-integer derivatives [10,11,12,13,14,15].

Fractional models are popular because, with a relatively small number of parameters, they can accurately describe the dynamic behavior of the damper for different temperatures and different frequencies. The small number of parameters used in the model is more advantageous because their identification based on experimental research is easier. For two-parameter models (e.g., the classical Kelvin or Maxwell model) some formulas can be derived that directly describe the parameters of the model (see, for example, [16]). Chang and Singh in [17] gave formulas to calculate the parameters of selected models to provide desired frequency-dependent characteristics represented by the storage and loss moduli. Lewandowski and Chorążyczewski [18] presented several methods of identifying the parameters of fractional models, including the use of hysteresis loops. Xu et al. in [19] proposed an equivalent fractional Kelvin model, which allows the simultaneous consideration of the effects of temperature and frequency. Fan et al. [20] used the Bayesian method to estimate the parameters of the generalized fractional Zener model. Polymer damping materials were characterized by a five-parameter fractional Zener model by Pritz in [21], where the high-frequency data was determined by the composite beam method for a wide temperature range. Lewandowski et al. in [22] derived the model parameters as a solution to a properly defined optimization problem, and to solve it, they used the particle swarm optimization method (PSO). In this work as well, an optimization problem was formulated with a slight modification, and the PSO method was used to determine the best values of the model parameters.

For viscoelastic materials that belong to the group of thermo-rheologically simple, the rheological properties (e.g., storage and loss modules) have the same functions in the appropriate frequency ranges, but for different temperatures. Therefore, it can be assumed that the function determined in the temperature *T*, after modifying the frequency range with the shift factor, will be valid for the reference temperature $T_{0}$. For these materials, it is possible to create the so-called master curves by shifting along the logarithmic frequency scale, the data obtained at different temperatures [23,24]. In some cases, such a shift should be made not only horizontally, but also vertically. In this way, rheological parameter functions (master curve) can be obtained over a wide range of frequencies, which are difficult or even impossible to obtain in laboratory conditions. The most popular method of determining the horizontal shift factor is the use of the Williams–Landel–Ferry (WLF) formula [23], in which there are certain constants that can be determined empirically. This empirical approach [25], or so-called “hand shifting”, requires some experience and does not guarantee that, for a given set of experimental data, each researcher will receive the same master curve, but it is a relatively easy and quick procedure. There are also numerical procedures for calculating the shift factor, based on the least squares method [26]. Gergesova et al. in [27] proposed a mathematical methodology in a closed form for the derivation of the shift algorithm, which avoids the uncertainties related to a manual shifting procedure.

This paper presents the method of determining the master curve for a selected viscoelastic material. The method of identifying the parameters of the Zener fractional model is also presented. The aim of this research is to accurately describe the rheological properties of the material for which the master curve was determined, for the selected temperature, in a given wide frequency range. There are no studies in the literature trying to determine the parameters of the rheological model on the basis of the master curve.

An innovative method of preparing a sample for the shear test is also proposed, in which four polymer layers are connected to four steel plates. Such an arrangement enables the axial fixing of the sample in the testing machine, but in this case it is necessary to take into account the distribution of forces and displacements in the sample.

Shear test specimens are commonly used in which the viscoelastic material is glued or vulcanized to steel elements. As the quality of such a connection may affect the test results, the paper proposes a special holder for shear tests, in which the polymer layer does not have to be glued to the steel sheets.

Following the description in Section 2.1 of the damper models used in this article and the equations of motion describing them, the corresponding complex modulus functions are given. The method of approximating the experimental results with continuous harmonic functions and the method of identifying the parameters of the adopted rheological model are described in Section 2.2. Section 2.3 discusses the influence of temperature on the dynamic behavior of the VE material, which was classified as thermorheologically simple. Section 3.1 presents the results of various parametric analyzes carried out in order to illustrate the impact of individual model parameters on the shape of module functions in a wide frequency range, i.e., master curves. Some conclusions regarding the effectiveness of the applied viscoelastic damper model for the correct description of their behavior depending on temperature and frequency are presented in Section 4.

## 2. Materials and Methods

### 2.1. Description of the Viscoelastic Material Properties

The dynamic behavior of a viscoelastic material can be described in various ways. Usually, the characteristics of a viscoelastic material are determined on the basis of laboratory tests. During the test, vibrations are forced in the material sample at a specified frequency, then the displacements and loads are measured in successive time steps.

#### 2.1.1. Hysteresis Loop

In order to create a hysteresis loop, a sinusoidal alternating strain ϵ(t) is applied to the viscoelastic material, and the response, the stress σ(t) is measured. For a viscoelastic material, the response is a function dependent on the excitation frequency λ and it is assumed that the response is delayed in relation to the excitation impulse by the phase angle δ:
(1)$$\begin{array}{c} \epsilon \left(t\right)=\epsilon_{0}sin\left(\lambda t\right)\\ \sigma \left(t\right)=\sigma_{0}sin\left(\lambda t+\delta \right) \end{array}$$
where, $\epsilon_{0}$ and $\sigma_{0}$ are the strain and stress amplitudes, respectively. The formula for stress, after some transformation, can be expressed as follows [3]:(2)$$\sigma \left(t\right)=\epsilon_{0}\left[E^{\prime }\left(\lambda \right)sin\left(\lambda t\right)+E^{\prime \prime }\left(\lambda \right)cos\left(\lambda t\right)\right]$$
where, $E^{\prime }\left(\lambda \right)=\sigma_{0}/\epsilon_{0}cos\left(\delta \right)$ is the so-called storage modulus, $E^{\prime \prime }\left(\lambda \right)=\sigma_{0}/\epsilon_{0}sin\left(\delta \right)$ is the loss modulus and the ratio between them is called the loss factor:(3)$$\eta =E^{\prime \prime }/E^{\prime }=tan\left(\delta \right)$$

After determining the functions of sin(λt) and cos(λt), respectively, from Equations (1) and (2), and using the relationship: $sin^{2}\left(\lambda t\right)+cos^{2}\left(\lambda t\right)=1$, the hysteresis loop function is obtained:(4)$${\left(\frac{\sigma \left(t\right)-E^{\prime }\left(\lambda \right)\epsilon \left(t\right)}{E^{\prime \prime }\left(\lambda \right)\epsilon_{0}}\right)}^{2}+{\left(\frac{\epsilon (t)}{\epsilon_{0}}\right)}^{2}=1$$

Figure 1 shows an example of the relationship between stress and strain, the so-called hysteresis loop for the VE material, where the previously mentioned values are marked: strain amplitude $\epsilon_{0}$ and accompanying stress $\sigma_{1}$, as well as stress amplitude $\sigma_{0}$ and accompanying strain $\epsilon_{3}$.

Moreover, the symbol $\sigma_{2}$ denotes the stress value, which corresponds to a strain equal to zero.

#### 2.1.2. Rheological Model of VE Material

The dynamic behavior of a VE material is often described using rheological models, e.g., the Kelvin model, the Maxwell model or the Zener model, which can be represented as a number of suitable combined springs and dashpots. However, in order to adequately describe the properties of the VE material over a wide range of temperatures and frequencies, the models need to be more complex and the number of parameters used to describe them must be large enough. Figure 2 shows complex rheological models, from which, after adopting appropriate parameters equal to zero, we can create several simple, classical models, e.g., for $E_{1}=c_{1}=0$ we get the Kelvin model, and for $E_{0}=c_{0}=0$ the Maxwell model is obtained.

Moreover, in the case where only $c_{0}=0$, the complex model of VE material simplifies to the Zener model. The physical equation of the Zener material model can be written as in [28]:(5)$$\sigma \left(t\right)+\tau \dot{\sigma }\left(t\right)=E_{0}\epsilon \left(t\right)+\tau E_{\infty }\dot{\epsilon }\left(t\right)$$
where, $E_{0}$ is treated as the relaxed stiffness modulus, $E_{\infty }=E_{0}+E_{1}$ is the non-relaxed stiffness and $\tau =c_{1}/E_{1}$ is the relaxation time for considered VE material. In the mechanical model shown in Figure 2a, the dashpot described by the parameter $c_{0}$ can be replaced by the Scott–Blair element, which is usually given in a form of rhombus (see Figure 2b) and is described by the two parameters $c_{0},\alpha$. The constitutive equation for Scott–Blair element could be written in the following way:(6)$$\sigma_{c}\left(t\right)=c_{0}D_{t}^{\alpha }\epsilon \left(t\right)$$
where, $D_{t}^{\alpha }\left(*\right)$ is the Riemann–Liouville derivative of the non-integer order (0<α≤1) with respect to time *t*, i.e., $D_{t}^{\alpha }\epsilon \left(t\right)=d^{\alpha }\epsilon \left(t\right)/dt^{\alpha }$ More information on non-integer derivatives can be found in [29]. After applying fractional derivatives, any classical rheological model changes into the corresponding fractional one. On the other hand, solutions for classical models can be obtained from solutions for fractional models after substituting α=1.

The equation of motion for the fractional Kelvin model and the fractional Maxwell model can be written for stresses and strains in a similar way as given in [18] for forces and displacements:(7)$$\begin{array}{c} \sigma \left(t\right)=E_{0}\epsilon \left(t\right)+E_{0}\tau_{K}^{\alpha }D_{t}^{\alpha }\epsilon \left(t\right)\\ \sigma \left(t\right)+\tau_{M}^{\alpha }D_{t}^{\alpha }\sigma \left(t\right)=E_{1}\tau_{M}^{\alpha }D_{t}^{\alpha }\epsilon \left(t\right) \end{array}$$
where, $\tau_{K}^{\alpha }=c_{0}^{\alpha }/E_{0}=c/E_{0},\;\tau_{M}^{\alpha }=c_{1}^{\alpha }/E_{1}=c/E_{1}$. The fractional Zener model, which can be called the four-parameter model $\left(E_{0},E_{1},c_{1},\alpha \right)$, consists of an elastic element (spring) connected in parallel with a fractional Maxwell element (see Figure 2b). When deriving the equation of motion for the Zener model, as for the Maxwell model, one should take into account the internal variable in the node connecting two elements, and then, after its elimination, one differential equation is obtained [22]:(8)$$\sigma \left(t\right)+\tau_{M}^{\alpha }D_{t}^{\alpha }\sigma \left(t\right)=E_{0}\epsilon \left(t\right)+E_{\infty }\tau_{M}^{\alpha }D_{t}^{\alpha }\epsilon \left(t\right)$$

The symbols $\tau_{M}^{\alpha }$ and $E_{\infty }$ used in the above formula are explained at Equations (5) and (7).

#### 2.1.3. Complex Modulus

Solution of the equation of motion for steady-state, harmonically excited vibrations $\epsilon \left(t\right)=\epsilon_{0}exp\left(i\lambda t\right)$, for the fractional model of VE material, takes the form $\sigma \left(t\right)=\sigma_{0}exp\left(i\lambda t\right)$, where $i=\sqrt{-1}$ is the imaginary unit. Next, assuming that λ>0 and that:(9)$$D_{t}^{\alpha }exp\left(i\lambda t\right)={\left(i\lambda \right)}^{\alpha }exp\left(i\lambda t\right)$$
one can derive the equation of motion (7) for the fractional Kelvin and fractional Maxwell models, respectively, as:(10)$$\sigma_{0}=E_{0}\left[1+{\left(i\tau_{K}\lambda \right)}^{\alpha }\right]\epsilon_{0};\;\sigma_{0}=E_{1}\frac{{\left(i\tau_{M}\lambda \right)}^{\alpha }}{1+{\left(i\tau_{M}\lambda \right)}^{\alpha }}\epsilon_{0}$$

Further, taking into account that $i^{\alpha }=cos\left(\alpha \pi /2\right)+i\cdot sin\left(\alpha \pi /2\right)$, the following expressions are obtained:(11)$$\begin{array}{c} \sigma_{0}=E_{0}\left[1+{\left(\tau_{K}\lambda \right)}^{\alpha }cos\left(\alpha \pi /2\right)+i\cdot {\left(\tau_{K}\lambda \right)}^{\alpha }sin\left(\alpha \pi /2\right)\right]\epsilon_{0}\\ \sigma_{0}=E_{1}{\left(\tau_{M}\lambda \right)}^{\alpha }\left[\frac{{\left(\tau_{M}\lambda \right)}^{\alpha }+cos\left(\alpha \pi /2\right)+i\cdot sin\left(\alpha \pi /2\right)}{1+{\left(\tau_{M}\lambda \right)}^{2}\alpha +2{\left(\tau_{M}\lambda \right)}^{\alpha }cos\left(\alpha \pi /2\right)}\right]\epsilon_{0} \end{array}$$

The constitutive relationship for a viscoelastic material that combines the normal stress σ(t) and the axial strain ϵ(t) can be expressed by the complex modulus $E^{*}\left(\lambda \right)$:(12)$$\begin{array}{c} \sigma \left(t\right)=E^{*}\left(\lambda \right)\epsilon \left(t\right)=\left[E^{\prime }\left(\lambda \right)+iE^{\prime \prime }\left(\lambda \right)\right]\epsilon \left(t\right)\\ \sigma \left(t\right)=E^{\prime }\left(\lambda \right)\left[1+i\eta \right]\epsilon \left(t\right) \end{array}$$

Comparing the Equations (11) and (12), the formulas for the storage modulus $E^{\prime }$ the loss modulus $E^{\prime \prime }$ and the loss factor η for the discussed material models are obtained [18]:for the fractional Kelvin model:(13)$$\begin{array}{c} E^{\prime }\left(\lambda \right)=E_{0}\left[1+{\left(\tau_{K}\lambda \right)}^{\alpha }cos\left(\alpha \pi /2\right)\right]\\ E^{\prime \prime }\left(\lambda \right)=E_{0}{\left(\tau_{K}\lambda \right)}^{\alpha }sin\left(\alpha \pi /2\right)\\ \eta =\frac{{\left(\tau_{K}\lambda \right)}^{\alpha }sin\left(\alpha \pi /2\right)}{1+{\left(\tau_{K}\lambda \right)}^{\alpha }cos\left(\alpha \pi /2\right)} \end{array}$$

for the fractional Maxwell model:


(14)
$$\begin{array}{c} E^{\prime }\left(\lambda \right)=E_{1}{\left(\tau_{M}\lambda \right)}^{\alpha }\frac{{\left(\tau_{M}\lambda \right)}^{\alpha }+cos\left(\alpha \pi /2\right)}{1+{\left(\tau_{M}\lambda \right)}^{2\alpha }+2{\left(\tau_{M}\lambda \right)}^{\alpha }cos\left(\alpha \pi /2\right)}\\ E^{\prime \prime }\left(\lambda \right)=E_{1}{\left(\tau_{M}\lambda \right)}^{\alpha }\frac{sin(\alpha \pi /2)}{1+{\left(\tau_{M}\lambda \right)}^{2\alpha }+2{\left(\tau_{M}\lambda \right)}^{\alpha }cos\left(\alpha \pi /2\right)}\\ \eta =\frac{sin(\alpha \pi /2)}{{\left(\tau_{M}\lambda \right)}^{\alpha }cos\left(\alpha \pi /2\right)} \end{array}$$


Corresponding solutions for classical Kelvin and Maxwell models can be obtained from Equations (13) and (14), after substituting α=1.

By performing the same transformations as described above, the elements of the complex modulus for the fractional Zener model [22] can be derived from Equation (Equation 8):(15)$$\begin{array}{c} E^{\prime }\left(\lambda \right)=\frac{E_{0}+\left(E_{0}+E_{\infty }\right){\left(\tau_{M}\lambda \right)}^{\alpha }cos\left(\alpha \pi /2\right)+E_{\infty }{\left(\tau_{M}\lambda \right)}^{2\alpha }}{1+{\left(\tau_{M}\lambda \right)}^{2\alpha }+2{\left(\tau_{M}\lambda \right)}^{\alpha }cos\left(\alpha \pi /2\right)}\\ E^{\prime \prime }\left(\lambda \right)=\frac{\left(E_{\infty }-E_{0}\right){\left(\tau_{M}\lambda \right)}^{\alpha }sin\left(\alpha \pi /2\right)}{1+{\left(\tau_{M}\lambda \right)}^{2\alpha }+2{\left(\tau_{M}\lambda \right)}^{\alpha }cos\left(\alpha \pi /2\right)}\\ \eta =\frac{\left(E_{\infty }-E_{0}\right){\left(\tau_{M}\lambda \right)}^{\alpha }sin\left(\alpha \pi /2\right)}{E_{0}+\left(E_{0}+E_{\infty }\right){\left(\tau_{M}\lambda \right)}^{\alpha }cos\left(\alpha \pi /2\right)+E_{\infty }{\left(\tau_{M}\lambda \right)}^{2\alpha }} \end{array}$$

It is worth noting that, as explained in Formula (7), $c_{1}^{\alpha }=c$, since $\tau_{M}^{\alpha }=c_{1}^{\alpha }/E_{1}=c/E_{1}$.

The complex modulus $E^{*}$ can be determined on the basis of the experimental data obtained in the compression-tensile test. Whereas, the associated complex, shear modulus:(16)$$G^{*}\left(\lambda \right)=G_{1}\left(\lambda \right)+iG_{2}\left(\lambda \right)$$
can be determined from the shear test data. The symbols $G_{1}\left(\lambda \right)$ and $G_{2}\left(\lambda \right)$ are the shear storage modulus and the shear loss modulus, respectively. In this case, the constitutive relationship connects the shear stress $\tilde{\tau }\left(t\right)$ with the shear deformation γ(t):(17)$$\tilde{\tau }\left(t\right)=G^{*}\left(\lambda \right)\gamma \left(t\right)$$

### 2.2. Identification Method

The selection of the method of parameter identification in the model of a viscoelastic material depends on the number of parameters used for its description. In classical models (Kelvin and Maxwell), there are only two parameters that must be determined, in the corresponding fractional models there are three parameters, and in the fractional Zener model there are four parameters.

Functional relationships will be determined below, first derived for the Zener model, and then for the set of data obtained from the experiment. They will then be compared to each other to identify the model parameters.

#### 2.2.1. Functional Relationships for the Zener Model

The dynamic behavior of a viscoelastic material can be described by a constitutive equation corresponding to the adopted material model. In the case of steady-state harmonic vibrations, the description of strains and stresses can be presented in the form of functions:(18)$$\epsilon \left(t\right)=\epsilon_{c}cos\left(\lambda t\right)+\epsilon_{s}sin\left(\lambda t\right);\;\sigma \left(t\right)=\sigma_{c}cos\left(\lambda t\right)+\sigma_{s}sin\left(\lambda t\right)$$

After substituting the function (18) to the equation of motion (8) for the Zener model and taking into account that:(19)$$D_{t}^{\alpha }cos\left(\lambda t\right)=\lambda^{\alpha }cos\left(\lambda t+\frac{\alpha \pi }{2}\right);\;D_{t}^{\alpha }sin\left(\lambda t\right)=\lambda^{\alpha }sin\left(\lambda t+\frac{\alpha \pi }{2}\right)$$
the following relationships can be derived:(20)$$\begin{array}{c} \sigma_{c}+{\left(\tau_{M}\lambda \right)}^{\alpha }\left(\sigma_{c}cos\frac{\alpha \pi }{2}+\sigma_{s}sin\frac{\alpha \pi }{2}\right)=E_{0}\epsilon_{c}+E_{\infty }{\left(\tau_{M}\lambda \right)}^{\alpha }\left(\epsilon_{c}cos\frac{\alpha \pi }{2}+\epsilon_{s}sin\frac{\alpha \pi }{2}\right)\\ \sigma_{s}+{\left(\tau_{M}\lambda \right)}^{\alpha }\left(\sigma_{s}cos\frac{\alpha \pi }{2}+\sigma_{c}sin\frac{\alpha \pi }{2}\right)=E_{0}\epsilon_{s}+E_{\infty }{\left(\tau_{M}\lambda \right)}^{\alpha }\left(\epsilon_{s}cos\frac{\alpha \pi }{2}+\epsilon_{c}sin\frac{\alpha \pi }{2}\right) \end{array}$$

The solution of the system of Equation (Equation 20) can be written in the following form:(21)$$\sigma_{c}=z_{1}\left(\lambda \right)\epsilon_{c}+z_{2}\left(\lambda \right)\epsilon_{s};\;\sigma_{s}=z_{1}\left(\lambda \right)\epsilon_{s}-z_{2}\left(\lambda \right)\epsilon_{c}$$
where,
(22)$$\begin{array}{c} z_{1}\left(\lambda \right)=\frac{E_{0}+\left(E_{0}+E_{\infty }\right){\left(\tau_{M}\lambda \right)}^{\alpha }cos\frac{\alpha \pi }{2}+E_{\infty }{\left(\tau_{M}\lambda \right)}^{2\alpha }}{1+{\left(\tau_{M}\lambda \right)}^{2\alpha }+2{\left(\tau_{M}\lambda \right)}^{\alpha }cos\frac{\alpha \pi }{2}}\\ z_{2}\left(\lambda \right)=\frac{\left(E_{\infty }-E_{0}\right){\left(\tau_{M}\lambda \right)}^{\alpha }sin\frac{\alpha \pi }{2}}{1+{\left(\tau_{M}\lambda \right)}^{2\alpha }+2{\left(\tau_{M}\lambda \right)}^{\alpha }cos\frac{\alpha \pi }{2}} \end{array}$$

Comparing the expression (15) describing the complex modulus for the Zener model with the expressions $z_{i}\left(\lambda \right)$ given in (22), we can conclude:(23)$$z_{1}\left(\lambda \right)\equiv E^{{}^{\prime }}\left(\lambda \right);\;z_{2}\left(\lambda \right)\equiv E^{\prime \prime }\left(\lambda \right)$$

The relationship (23) is valid for all material models discussed above.

#### 2.2.2. Experimental Data Approximation

Regardless of the adopted material model and the number of parameters present in the model, at the first stage of the identification process, discrete numerical data obtained from the experiment are approximated by continuous functions. During the experiment, the testing machine records the displacement $q_{e}\left(t_{j}\right)$ and the corresponding force $U_{e}\left(t_{j}\right)$ at a specific time instant $t_{j}$ and then at specified time intervals $▵t=t_{j+1}-t_{j}$, i.e., with a given sampling frequency $f_{s}=1/▵t$. On the basis of the geometrical dimensions of the test specimen, strains and stresses can be determined.

The set of data obtained from the measurement is approximated by the continuous harmonic function, which for displacements has the form:(24)$$\tilde{q}\left(t\right)={\tilde{q}}_{c}cos\left(\lambda t\right)+{\tilde{q}}_{s}sin\left(\lambda t\right)$$
and for forces is assumed as:(25)$$\tilde{U}\left(t\right)={\tilde{u}}_{c}cos\left(\lambda t\right)+{\tilde{u}}_{s}sin\left(\lambda t\right)$$
where, ${\tilde{q}}_{c},{\tilde{q}}_{s},{\tilde{u}}_{c},{\tilde{u}}_{s}$ are the amplitudes for displacements and forces, respectively, the values of which should be determined for the given excitation frequency λ. For this purpose, the least squares method was used [22] and the following functional is minimized:(26)$$J_{1}=\frac{1}{t_{2}-t_{1}}\int_{t_{1}}^{t_{2}}{\left[q_{e}\left(t\right)-\tilde{q}\left(t\right)\right]}^{2}dt$$
where $t_{1}$ and $t_{2}$ are the limits of the time interval in which the displacement measurements were recorded. From the stationary conditions of the functional (26), the following system of equations can be written:(27)$$\begin{array}{c} I_{cc}{\tilde{q}}_{c}+I_{cs}{\tilde{q}}_{s}=I_{cq}\\ I_{sc}{\tilde{q}}_{c}+I_{ss}{\tilde{q}}_{s}=I_{sq} \end{array}$$
where
(28)$$\begin{array}{c} I_{cc}=\int_{t_{1}}^{t_{2}}cos^{2}\left(\lambda t\right)dt\\ I_{sc}=I_{cs}=\int_{t_{1}}^{t_{2}}sin\left(\lambda t\right)\cdot cos\left(\lambda t\right)dt\\ I_{ss}=\int_{t_{1}}^{t_{2}}sin^{2}\left(\lambda t\right)dt \end{array}$$

Since the measured values of $q_{e}\left(t\right)$ have a discrete distribution, the terms on the right-hand side of the system of Equation (Equation 27) are calculated as the sums of successive integrals:(29)$$\begin{array}{c} I_{cq}=\int_{t_{1}}^{t_{2}}q_{e}\left(t\right)\cdot cos\left(\lambda t\right)dt=\sum_{j=1}^{N}q_{e}\left(t_{j}\right)\cdot ▵t\int_{t_{j}}^{t_{j+1}}cos\left(\lambda t\right)dt\\ I_{sq}=\int_{t_{1}}^{t_{2}}q_{e}\left(t\right)\cdot sin\left(\lambda t\right)dt=\sum_{j=1}^{N}q_{e}\left(t_{j}\right)\cdot ▵t\int_{t_{j}}^{t_{j+1}}sin\left(\lambda t\right)dt \end{array}$$
where $▵t=t_{j+1}-t_{j}$ depends on the sampling frequency, i.e., $▵t=1/f_{s}$ and $N=(t_{2}-t_{1})/▵t$. After solving the system of Equation (Equation 27), we obtain the desired amplitudes of the function approximating the set of recorded displacements, i.e., ${\tilde{q}}_{c}$ and ${\tilde{q}}_{s}$.

The amplitudes of the function approximating the set of recorded forces (${\tilde{u}}_{c}$ and ${\tilde{u}}_{s}$) can be determined after introducing and replacing the appropriate quantities in the system of Equation (Equation 27), i.e.,
(30)$$\begin{array}{c} I_{cc}{\tilde{u}}_{c}+I_{cs}{\tilde{u}}_{s}=I_{cu}\\ I_{sc}{\tilde{u}}_{c}+I_{ss}{\tilde{u}}_{s}=I_{su} \end{array}$$
where
(31)$$\begin{array}{c} I_{cu}=\int_{t_{1}}^{t_{2}}u_{e}\left(t\right)\cdot cos\left(\lambda t\right)dt=\sum_{j=1}^{N}u_{e}\left(t_{j}\right)\cdot ▵t\int_{t_{j}}^{t_{j+1}}cos\left(\lambda t\right)dt\\ I_{su}=\int_{t_{1}}^{t_{2}}u_{e}\left(t\right)\cdot sin\left(\lambda t\right)dt=\sum_{j=1}^{N}u_{e}\left(t_{j}\right)\cdot ▵t\int_{t_{j}}^{t_{j+1}}sin\left(\lambda t\right)dt \end{array}$$

The other symbols used in Equation (Equation 31) are explained in Equation (Equation 29).

From the systems of Equations (27) and (30), it is possible to determine the amplitudes of the function approximating the force (${\tilde{u}}_{c}$ and ${\tilde{u}}_{s}$) and displacements (${\tilde{q}}_{c}$ and ${\tilde{q}}_{s}$), which will be used to identify the parameters of material models.

#### 2.2.3. Functional Relations for Measured Data

The displacements and forces measured in the experiment, as well as the corresponding strains and stresses, are in a certain relationship that can be expressed as:(32)$$\begin{array}{c} {\tilde{u}}_{c}={\tilde{\phi }}_{1}\left(\lambda \right){\tilde{q}}_{c}+{\tilde{\phi }}_{2}\left(\lambda \right){\tilde{q}}_{s}\\ {\tilde{u}}_{s}=-{\tilde{\phi }}_{2}\left(\lambda \right){\tilde{q}}_{c}+{\tilde{\phi }}_{1}\left(\lambda \right){\tilde{q}}_{s} \end{array}$$
where, ${\tilde{\phi }}_{1}\left(\lambda \right)$ and ${\tilde{\phi }}_{2}\left(\lambda \right)$ are functions that must be met for the data obtained from the experiment, and must satisfy the equations of motion of the adopted material model, for each excitation frequency λ.

The determined amplitudes of functions approximating forces and displacements can be easily used to determine the amplitudes of stresses and strains. Appropriate geometrical dimensions of the test specimen, i.e., B,L and *h*, are needed in the case of compressive or tensile test (see Figure 3):(33)$$\begin{array}{c} {\tilde{\sigma }}_{i}=\frac{{\tilde{u}}_{i}}{B\cdot L}\\ {\tilde{\epsilon }}_{i}=\frac{{\tilde{q}}_{i}}{h} \end{array}$$
as well as for the shear test, although they indicate different dimensions (see Figure 4):(34)$$\begin{array}{c} {\tilde{\tau }}_{i}=\frac{{\tilde{u}}_{i}}{B\cdot L}\\ {\tilde{\gamma }}_{i}=\frac{{\tilde{q}}_{i}}{h} \end{array}$$

The index “*i*” stands for “*c*” or “*s*” appearing in Equation (Equation 32). Figure 3 shows the above-mentioned dimensions of a specimen tested in a tensile or compression test, which are needed to determine the value of normal stress.

In addition, Figure 4 shows the required dimensions of the sample tested in the shear test, necessary to determine the value of the shear stress.

Using the relationships (33) in Equation (Equation 32), it is possible to write the expression that combines stresses and strains:(35)$$\begin{array}{c} {\tilde{\sigma }}_{c}=\phi_{1}\left(\lambda \right){\tilde{\epsilon }}_{c}+\phi_{2}\left(\lambda \right){\tilde{\epsilon }}_{s}\\ {\tilde{\sigma }}_{s}=-\phi_{2}\left(\lambda \right){\tilde{\epsilon }}_{c}+\phi_{1}\left(\lambda \right){\tilde{\epsilon }}_{s} \end{array}$$
where, $\phi_{1}\left(\lambda \right)=h{\tilde{\phi }}_{1}\left(\lambda \right)/BL$ and $\phi_{2}\left(\lambda \right)=h{\tilde{\phi }}_{2}\left(\lambda \right)/BL$. After transformations, from the system of Equation (Equation 35), the $\phi_{i}\left(\lambda \right)$ functions, which determine the relationship between strains and stresses obtained in the experiment, are derived:(36)$$\begin{array}{c} \phi_{1}\left(\lambda \right)=\frac{{\tilde{\sigma }}_{c}{\tilde{\epsilon }}_{c}+{\tilde{\sigma }}_{s}{\tilde{\epsilon }}_{s}}{\epsilon_{c}^{2}+\epsilon_{s}^{2}}\\ \phi_{2}\left(\lambda \right)=\frac{{\tilde{\sigma }}_{c}{\tilde{\epsilon }}_{s}-{\tilde{\sigma }}_{s}{\tilde{\epsilon }}_{c}}{\epsilon_{c}^{2}+\epsilon_{s}^{2}} \end{array}$$

Strain and stress amplitudes as well as $\phi_{i}$ functions should be determined for each analyzed excitation frequency λ. For shear stresses, the relations (36) are also valid. In this case, we have to substitute ${\tilde{\gamma }}_{c}$ and ${\tilde{\gamma }}_{s}$ for strains and ${\tilde{\tau }}_{c}$ and ${\tilde{\tau }}_{s}$ for stresses, according to the symbols in Equation (Equation 34).

#### 2.2.4. Identification of Model Parameters

In order to identify the parameters of the adopted model, e.g., $E_{0},E_{\infty },\tau_{M}$ and α for the Zener model, it is required that the differences between the relations $z_{i}\left(\lambda \right)$ in Equation (Equation 22) derived for the model and the relations $\phi_{i}\left(\lambda \right)$ given in Equation (Equation 36), which were derived for the data set obtained from the experiment, are minimal. The above-mentioned functions should be determined for each excitation frequency $\lambda_{r}$ for which the experiments were performed. In other words, we are looking for such values for the parameters of the analyzed model, at which the minimum is reached by the following functional [22]:(37)$$J_{2}=\sum_{j=1}^{R}\left[{\left(z_{1}\left(\lambda_{r}\right)-\phi_{1}\left(\lambda_{r}\right)\right)}^{2}+{\left(z_{2}\left(\lambda_{r}\right)-\phi_{2}\left(\lambda_{r}\right)\right)}^{2}\right]$$
where, *R* is the number of frequencies for which the experiments and the calculation of the relevant parameters were performed. Moreover, the model parameters have certain constraints. All parameters should be non-negative, and parameter α, as the order of the fractional derivative, should be in the range (0,1〉, e.g., for the Zener model we have:(38)$$\begin{array}{c} E_{\infty }>E_{0}>0\\ \tau_{M}>0\\ 0<\alpha \leq 1.0 \end{array}$$

If the scope of the tests is limited to one excitation frequency $\lambda_{1}$ and the classical Kelvin model (α=1.0) is used to describe the material, then the functional (37) can be limited to a system of two equations:(39)$$\begin{array}{c} z_{1}\left(\lambda_{1}\right)=\phi_{1}\left(\lambda_{1}\right)\to E_{0}\left(\lambda_{1}\right)=\frac{{\tilde{\sigma }}_{c}{\tilde{\epsilon }}_{c}+{\tilde{\sigma }}_{s}{\tilde{\epsilon }}_{s}}{\epsilon_{c}^{2}+\epsilon_{s}^{2}}\\ z_{2}\left(\lambda_{1}\right)=\phi_{2}\left(\lambda_{1}\right)\to c_{0}\left(\lambda_{1}\right)=\frac{{\tilde{\sigma }}_{c}{\tilde{\epsilon }}_{s}-{\tilde{\sigma }}_{s}{\tilde{\epsilon }}_{c}}{\lambda_{1}\left(\epsilon_{c}^{2}+\epsilon_{s}^{2}\right)} \end{array}$$

In the case of models described with a greater number of parameters, in order to determine the functional minimum (37), one of the multi-parameter optimization methods should be used. One of such methods is the Particle Swarm Optimization (PSO) method, which is a gradientless method consisting in searching the domain of possible solutions [30].

In the PSO method, the number of elements $N_{p}$ is first determined, i.e., the number of the so-called particles for which the objective function value is calculated at each iteration step. The location of the particle is determined by the vector whose components are the parameters of the adopted material model, e.g., for the Zener model the position vector consists of four elements, $p_{e}={\left[E_{0},E_{\infty },\tau_{M},\alpha \right]}^{T}$, where $e=1,\ldots ,N_{p}$. In the first iteration step (k), the components of the position vector are assumed as random numbers that must satisfy the imposed constraints, given in Equation (Equation 38). Then, for each particle, the value of the objective function is calculated, and the particle for which the functional (37) has the minimal value is selected. The position vector of the particle selected in this way is treated as the best position g(k=1) so far found for the whole considered swarm of particles. In the next iteration step (k+1), the velocity vector is determined, which aims to direct individual particles towards the best position:(40)$$v_{e}\left(k+1\right)=w\left(k\right)v_{e}\left(k\right)+\frac{{\overset{˘}{C}}_{1}}{▵t}R_{1}\left(k\right)\left(b_{e}\left(k\right)-p_{e}\left(k\right)\right)+\frac{{\overset{˘}{C}}_{2}}{▵t}R_{2}\left(k\right)\left(g\left(k\right)-p_{e}\left(k\right)\right)$$
where, ${\overset{˘}{C}}_{1}$ and ${\overset{˘}{C}}_{2}$ are arbitrarily chosen constants to control the convergence process (e.g., ${\overset{˘}{C}}_{1}={\overset{˘}{C}}_{2}=2.0$), ▵t is the assumed time step (e.g., ▵t=1.0 s), $R_{1}\left(k\right)$ and $R_{2}\left(k\right)$ are the diagonal matrices of random numbers, uniformly distributed in the range (0,1), $b_{e}\left(k\right)$ is the vector of the best position for a given particle *e* found so far, but in the second iteration step $b_{e}\left(k=1\right)=p_{e}\left(1\right)$ should be assumed. The velocity vector in the first step of iteration can be taken equal to zero, i.e., $v_{e}\left(k=1\right)=0$, while w(k) is a weighting factor that may change in subsequent steps, e.g., decrease linearly [30]. The new position of the particles is determined taking into account the coordinates of the velocity vector:(41)$$p_{e}\left(k+1\right)=p_{e}\left(k\right)+v_{e}\left(k+1\right)▵t$$

The iterative procedure can be interrupted when the change of the best value of the objective function is relatively small after several consecutive steps equal to $n_{s}$, i.e., when:(42)$$|J_{2}\left(k+n_{s}\right)-J_{2}\left(k\right)|\leq \epsilon_{1}\cdot J_{2}\left(k+n_{s}\right)$$
where, $\epsilon_{1}$ is sufficiently small number, e.g., $\epsilon_{1}=0.05$.

### 2.3. Temperature Influence

To determine the influence of temperature on the dynamic behavior of VE material, the principle of temperature–frequency superposition was used [31]. According to this principle, for a thermorheologically simple material, the value of the complex modulus $E^{*}$ determined for the selected frequency and temperature (λ,T) is equal to the value of the modulus for the adequate frequency at reference temperature $\left(\lambda_{0},T_{0}\right)$. Moreover, this adequate frequency value at the reference temperature can be derived using the shift factor $\alpha_{T}$ (i.e., $\lambda_{0}=\alpha_{T}\lambda$):(43)$$E^{*}\left(\lambda ,T\right)=E^{*}\left(\lambda_{0},T_{0}\right)=E^{*}\left(\alpha_{T},\lambda ,T_{0}\right)$$

Such a shift in the frequency domain of the data obtained for different temperatures allows for the creation of a functional relationship for a complex modulus, the so-called master curve for a wide frequency range.

The horizontal shift factor $\alpha_{T}$ is usually determined empirically. In this work, we use the formula proposed by William–Landel–Ferry [32]:(44)$$log\;\alpha_{T}=\frac{-C_{1}▵T}{C_{2}+▵T}$$
where, $C_{1}$ and $C_{2}$ are the experimentally determined constants and ▵T is the difference between the actual temperature and the reference temperature $▵T=T-T_{0}$. For the fractional Zener model, the storage and the loss modules determined for the reference frequency $\lambda_{0}$ and at the reference temperature $T_{0}$ are described by the equations:(45)$$\begin{array}{c} E^{{}^{\prime }}\left(\lambda_{0},T_{0}\right)=\frac{E_{0}+\left(E_{0}+E_{\infty }\right){\left(\tau_{M}\lambda_{0}\right)}^{\alpha }cos\left(\alpha \pi /2\right)+E_{\infty }{\left(\tau_{M}\lambda_{0}\right)}^{2\alpha }}{1+{\left(\tau_{M}\lambda_{0}\right)}^{2\alpha }+2{\left(\tau_{M}\lambda_{0}\right)}^{\alpha }cos\left(\alpha \pi /2\right)}\\ E^{\prime \prime }\left(\lambda_{0},T_{0}\right)=\frac{\left(E_{0}-E_{\infty }\right){\left(\tau_{M}\lambda_{0}\right)}^{\alpha }sin\left(\alpha \pi /2\right)}{1+{\left(\tau_{M}\lambda_{0}\right)}^{2\alpha }+2{\left(\tau_{M}\lambda_{0}\right)}^{\alpha }cos\left(\alpha \pi /2\right)} \end{array}$$

For the actual frequency λ and actual temperature *T*, the above functions can be given as follows:(46)$$\begin{array}{c} E^{{}^{\prime }}\left(\lambda ,T\right)=\frac{{\tilde{E}}_{0}+\left({\tilde{E}}_{0}+{\tilde{E}}_{\infty }\right){\left({\tilde{\tau }}_{M}\lambda_{0}\right)}^{\alpha }cos\left(\alpha \pi /2\right)+{\tilde{E}}_{\infty }{\left({\tilde{\tau }}_{M}\lambda_{0}\right)}^{2\alpha }}{1+{\left({\tilde{\tau }}_{M}\lambda_{0}\right)}^{2\alpha }+2{\left({\tilde{\tau }}_{M}\lambda_{0}\right)}^{\alpha }cos\left(\alpha \pi /2\right)}\\ E^{\prime \prime }\left(\lambda ,T\right)=\frac{\left({\tilde{E}}_{0}-{\tilde{E}}_{\infty }\right){\left({\tilde{\tau }}_{M}\lambda_{0}\right)}^{\alpha }sin\left(\alpha \pi /2\right)}{1+{\left({\tilde{\tau }}_{M}\lambda_{0}\right)}^{2\alpha }+2{\left({\tilde{\tau }}_{M}\lambda_{0}\right)}^{\alpha }cos\left(\alpha \pi /2\right)} \end{array}$$

However, taking into account the relationship (43), the functions of the modules (46) are equivalent to the solutions obtained at the reference temperature $T_{0}$ and the adjusted frequency $\lambda_{0}=\alpha_{T}\lambda$:(47)$$\begin{array}{c} E^{{}^{\prime }}\left(\alpha_{T},\lambda ,T_{0}\right)=\frac{E_{0}+\left(E_{0}+E_{\infty }\right){\left(\tau_{M}\alpha_{T}\lambda \right)}^{\alpha }cos\left(\alpha \pi /2\right)+E_{\infty }{\left(\tau_{M}\alpha_{T}\lambda \right)}^{2\alpha }}{1+{\left(\tau_{M}\alpha_{T}\lambda \right)}^{2\alpha }+2{\left(\tau_{M}\alpha_{T}\lambda \right)}^{\alpha }cos\left(\alpha \pi /2\right)}\\ E^{\prime \prime }\left(\alpha_{T},\lambda ,T_{0}\right)=\frac{\left(E_{0}-E_{\infty }\right){\left(\tau_{M}\alpha_{T}\lambda \right)}^{\alpha }sin\left(\alpha \pi /2\right)}{1+{\left(\tau_{M}\alpha_{T}\lambda \right)}^{2\alpha }+2{\left(\tau_{M}\alpha_{T}\lambda \right)}^{\alpha }cos\left(\alpha \pi /2\right)} \end{array}$$

By comparing solutions (46) and (47), the following relationships can be formulated:(48)$$\begin{array}{c} {\tilde{E}}_{0}=E_{0}\\ {\tilde{E}}_{\infty }=E_{\infty }\\ {\tilde{\tau }}_{M}=\alpha_{T}\tau_{M} \end{array}$$

The above analyzes show that for a thermoreologically simple VE material, the influence of temperature change can be taken into account also by appropriate modification of the damping parameter in the model under consideration, because $\alpha_{T}\tau_{M}=\alpha_{T}C_{1}/E_{1}$ [32].

## 3. Results and Discussion

### 3.1. Test Results for Selected VE Materials

A material with viscoelastic properties, i.e., polyurethane with the trade name B-85, was selected for laboratory tests. It is a material mainly used to reduce vibrations, the manufacturer of which (Boral) declares the following mechanical properties: hardness 89 ShA, tensile strength 31 MPa, compression set (70 ${}^{\circ }$C/24 h) 15% and rebound 43%.

In order to confirm the isotropy of the material, a series of static compression tests were performed. Cubic samples were cut from the viscoelastic material layer (Figure 5), and then three samples were compressed in the X direction (in the layer plane), three in the Y direction (also in the layer plane) and three in the Z direction (perpendicular to the layer plan).

The results obtained from compressing the material in three perpendicular directions were very similar. Figure 6 shows the change in force and displacement recorded until the sample was completely crushed. The samples with dimensions of 26 × 26 × 26 mm were compressed by almost 20 mm. Initially, with a displacement of several millimeters, the process was linear. The stresses and strains were determined according to Equation (Equation 33), and then the average value of Young’s modulus for the tested material was determined as *E* = 18,929 [kPa], regardless of the direction.

The parameter identification procedure was based on the results of laboratory tests carried out in two different shear tests. In the first approach, a sample of the viscoelastic material was prepared with a notch and subjected to shear in a weakened section, i.e., a transverse shear test. In the second approach, four layers of VE material were glued to metal sheets and sheared along the longest dimension of the specimen, i.e., longitudinal shear test.

#### 3.1.1. Transverse Shear Test

A sample of the layer of viscoelastic material in the center was cut from the top and bottom, thus creating a weakened cross-section with reduced area (Figure 7). An exemplary sample shown in Figure 7 has the width B=51.4 mm, the thickness at the notch point L=13.3 mm, and the width of the slot h=4.8 mm (dimensions B,L and *h* are given in accordance with Figure 4). In this case, the area of the shear plane is B×L=683.62 mm${}^{2}$.

Additionally, in order to carry out the shear test, a special holder was built, which allows forcing shear in the weakened section (Figure 8). During the experiment, a transverse displacement was imposed on the sample, while displacements and forces were measured simultaneously. The tests were performed for several different forced frequencies, and the sample was subjected to several cycles for each frequency.

Figure 9 shows the measurement results for the forcing frequency f=22.29 Hz (λ=140 rad/s) in the form of a hysteresis loop (points). After using the Formulas (27) and (30), the amplitudes of the approximating functions were determined. These amplitudes allowed us to plot the hysteresis loop in the form of a continuous function (solid line in Figure 9). Then, from the system of Equation (Equation 32), functions ${\tilde{\phi }}_{i}\left(\lambda \right)$ were determined, which are relations between displacements ${\tilde{q}}_{i}$ and forces ${\tilde{u}}_{i}$ for a given excitation frequency λ.

Table 1 lists the values of the amplitudes of the functions approximating displacements and forces, as well as the values of the ${\tilde{\phi }}_{i}$ functions expressing the relationships between them, determined for the excitation frequency λ=140 rad/s. The appropriate amplitudes of stresses and strains, calculated from the relationships (34) and the values of the functions $\phi_{i}$, denoting the relationships between them, and calculated from the Formula (36) are summarized in Table 2.

In a similar way, measurements were carried out for a set of selected frequencies from 1.0 rad/s to 150 rad/s. After determining the amplitudes of the approximating functions, i.e., ${\tilde{\gamma }}_{c}\left(\lambda_{i}\right),{\tilde{\gamma }}_{s}\left(\lambda_{i}\right),{\tilde{\tau }}_{c}\left(\lambda_{i}\right),{\tilde{\tau }}_{s}\left(\lambda_{i}\right)$, the functional relationships $\phi_{1}\left(\lambda_{i}\right)$ (dots in Figure 10) and $\phi_{2}\left(\lambda_{i}\right)$ (dots in Figure 11) were determined. Then, using the PSO method, the parameters of the Zener model were determined: $E_{0}=1930$ [kPa], $E_{\infty }=3737$ [kPa], $\tau_{M}=0.0014$ and α=0.47. Figure 10 and Figure 11 show the results, the values of the functions $z_{1}\left(\lambda \right)\equiv G_{1}\left(\lambda \right)$ and $z_{2}\left(\lambda \right)\equiv G_{2}\left(\lambda \right)$, respectively, calculated for the above-determined model parameters (solid lines in Figure 10 and Figure 11).

It is worth noting that the PSO method, as a method of searching through a set of possible solutions, requires multiple startups and selecting the best solution. Sometimes the values of the control parameters (${\overset{˘}{C}}_{1},{\overset{˘}{C}}_{2}$ and w(k)) must be changed so that the swarm of particles is properly dispersed and searches a sufficiently large area.

In Figure 10 and Figure 11, can be observed a good convergence of the results obtained from the experiment and the solutions obtained for the Zener model, in which the determined model parameters were used.

#### 3.1.2. Longitudinal Shear Test

A sample for longitudinal shear tests was prepared by gluing four layers of viscoelastic material to four steel plates (Figure 12). The adopted method of connecting four plates and four viscoelastic layers allows for axial installation and loading of the system in a testing machine and performing a shear test.

The set shown in Figure 12 consists of four identical viscoelastic layers with a width $B_{1}=100$ mm, length L=70 mm and thickness $h_{1}=9$ mm (dimensions $B_{1},L$ and $h_{1}$ correspond to those shown in Figure 4).

Due to the complex structure of the entire sample and the number of layers used, the method of calculating stresses should be analyzed in more detail. As explained in Figure 13, it is assumed that the force acting on the system is distributed over two polymer layers and the displacement amplitude is reduced twice in one layer.

Assuming that, in this case, the stresses and strains are calculated for reduced force and displacement amplitudes:(49)$$\begin{array}{c} {\tilde{\tau }}_{i}=\frac{\frac{{\tilde{u}}_{i}}{2}}{B_{1}\cdot L}=\frac{{\tilde{u}}_{i}}{2B_{1}\cdot L}\\ {\tilde{\gamma }}_{i}=\frac{\frac{{\tilde{q}}_{i}}{2}}{h_{1}}=\frac{{\tilde{q}}_{i}}{2h_{1}} \end{array}$$
it can be written that:(50)$$\begin{array}{c} B=2B_{1}\\ h=2h_{1} \end{array}$$
which leads to the formulas for functions that combine stresses and strains:(51)$$\phi_{i}\left(\lambda \right)=\frac{h}{B\cdot L}{\tilde{\phi }}_{i}\left(\lambda \right)=\frac{h_{1}}{B_{1}\cdot L}{\tilde{\phi }}_{i}\left(\lambda \right)$$
where, i=1,2.

In the first stage, the sample was cooled to −30
${}^{\circ }$C, and then excited with selected frequencies ranging from 0.1 Hz to 4 Hz. The sample was then heated up to 70 ${}^{\circ }$C with a heating rate of 2 ${}^{\circ }$C/min and, sequentially, when the temperature increased by another 10 degrees, the dynamic test was repeated.

The results obtained from the measurements were approximated by the functions (24) and (25), and on their basis, the relationships $\phi_{1}\left(\lambda_{i}\right)$ and $\phi_{2}\left(\lambda_{i}\right)$ were determined. The functions $\phi_{i}\left(\lambda_{i}\right)$ are equivalent to the shear storage modulus and the shear loss modulus, respectively, i.e., $G_{1}\left(\lambda_{i}\right)\equiv \phi_{1}\left(\lambda_{i}\right)$ and $G_{2}\left(\lambda_{i}\right)\equiv \phi_{2}\left(\lambda_{i}\right)$.

#### 3.1.3. Construction of a Master Curve Diagram

The validity of the time-temperature superposition principle for the tested material was checked by creating the Cole–Cole plot (Figure 14) and the wicket plot (Figure 15) [33,34].

Figure 15 shows the variation in the tangent of the phase angle, which is defined in the same way as the loss factor:(52)$$\eta =\frac{G_{2}\left(\lambda \right)}{G_{1}\left(\lambda \right)}=tan\left(\delta \right)$$

In order to determine the shift factor $\alpha_{T}$, a plot of the loss modulus for selected temperatures in the frequency domain was created (Figure 16). The value $T_{0}=0{\;}^{\circ }$C was chosen as the reference temperature. Figure 16 shows the points where the value of the loss modulus is the same for different temperatures, they correspond to different frequencies.

The temperature differences $▵T_{i}=T_{i}-T_{0}$ and the frequency ratios $\lambda_{i}/\lambda_{0}$ corresponding to these points allow determining the constants $C_{1}$ and $C_{2}$ from the Formula (44).

The determined constants $C_{1}=18.08$ and $C_{2}=135.74$ substituted into the Formula (44) lead to the shift factor $\alpha_{T}$ as function of the relative temperature $▵T_{i}$, the diagram of which is shown in Figure 17.

Then, using the values of the shift factor $\alpha_{T}$, the graphs of modules were created in a wide range of frequencies.

The so-called master curves were created for the shear storage modulus $G_{1}$ (points in Figure 18) and for the shear loss modulus $G_{2}$ (points in Figure 19).

Although there are some gaps between the measured values in the graphs, the general trend is visible.

The approximated results from the experiment were then used in the PSO procedure to identify the parameters of the fractional Zener model. Calculations were performed several times as described in Section 2.2.4, each time a different configuration of control parameters was adopted, i.e., the constants responsible for the convergence of the calculation process. Finally, the following parameters were selected as the best fit of the model to the experiment results: $E_{0}=1400$ [kPa], $E_{\infty }$ = 18,000 [kPa], $\tau_{M}=0.0001$ and α=0.27. In Figure 18 and Figure 19, the functions of the storage and loss modules determined for the Zener model and for the adopted parameters are shown as solid lines.

In order to assess the quality of fitting the function describing the model to empirical data, the mean square error was determined. By analyzing the entire frequency range, i.e., from $1.0\times 10^{-6}$ Hz to $1.0\times 10^{6}$ Hz, where the values of the modules vary in the range: $G_{1}=1.36÷15.78$ MPa and $G_{2}=0.07÷2.96$ MPa, mean square errors $S_{a1}=0.707$ and $S_{a2}=0.924$ were obtained, respectively, for each module. By narrowing the frequency domain to the range from $1.0\times 10^{-2}$ Hz to 1.0×10 Hz, where the module values vary in the range: $G_{1}=1.82÷4.07$ MPa and $G_{2}=0.12÷1.23$ MPa, mean square errors were obtained, respectively, $S_{r1}=0.407$ and $S_{r2}=0.075$.

The second measure of the correctness of the model was the mean relative percentage error, calculated as the ratio of the difference between the results of the experiment and the model to the value obtained from the experiment. For the entire frequency range, i.e., twelve decades, the errors $R_{a1}=8.18\%$ and $R_{a2}=40.25\%$ were obtained, respectively, for the $G_{1}$ and $G_{2}$ module. For the frequency range covering three decades, more even errors were obtained for both modules, i.e., $R_{r1}=10.75\%$ and $R_{r2}=15.37\%$.

Due to the fact that the loss modulus function $G_{2}\left(\lambda \right)$ calculated for the Zener model coincides with the experimental results only in the middle frequency range (Figure 19), further analyzes were carried out. For each of the four parameters of the fractional Zener model, i.e., $E_{0},E_{\infty },\tau_{M}$ and α, an analysis of the impact of changing the parameter value on the solution was performed. It is worth noting that the quantities appearing in the expressions for the modules (Equation (Equation 15)) and those given in Figure 2b are in the following relationships: $E_{1}=E_{\infty }-E_{0}$ and $c_{1}=\tau_{M}E_{1}$. Table 3 lists the values of the model parameters that were selected for additional calculations. For each model parameter, the value that was considered to be the best fit for the model was given, and one smaller and one larger value were selected.

Table 3 shows how the change in the value of the selected model parameter affects the average relative percentage error calculated for each of the modules.

Figure 20 and Figure 21 show the functions of the storage and loss modules for three different values of the $E_{0}$ parameter, the other parameters are unchanged, always those that best fit the model. On the basis of the obtained diagrams, it can be concluded that the change of the $E_{0}$ parameter value does not affect the $G_{2}$ modulus function and that it changes the $G_{1}$ modulus values only in the range of lower frequencies (higher temperatures).

Figure 22 and Figure 23 show the effect of changing the $E_{1}$ parameter value on the storage and loss modules. An increase in the value of the $E_{1}$ parameter causes a uniform increase in the value of the storage modulus with the increase in frequency and causes an equal increase in the loss modulus regardless of the frequency value.

Figure 24 and Figure 25 show how changing the value of parameter $c_{1}$ affects the storage and loss modules. Generally, changing the $c_{1}$ parameter (i.e., also $\tau_{M}=c_{1}/E_{1}$) causes a horizontal shift in the module graphs.

Finally, in Figure 26 and Figure 27, the storage and loss modules for different values of the α parameter are shown.

It can be concluded that reducing the value of the α parameter flattens the graphs of the modules $G_{1}$ and $G_{2}$.

The model parameters identification procedure was also used for Kelvin and Maxwell fractional models. In the PSO method for the Kelvin model, the parameters: $E_{0}=2180$ [kPa], $\tau_{K}=0.035$ and α=0.18 were obtained. On the other hand, for the Maxwell model, the best fit was obtained for the parameters: $E_{1}=$ 21,360 [kPa], $\tau_{M}=0.0001$ and α=0.22. Figure 28 and Figure 29 present the results of the experiment and the designated functions of the modules for the three-parameter Kelvin and Maxwell models, as well as for the four-parameter Zener model. The smaller number of parameters in the model (Kelvin, Maxwell) increased the discrepancy between the function of the storage module $G_{1}$ and the results of the experiments for lower frequencies.

## 4. Conclusions

The aim of the study was to determine the parameters of the fractional Zener model so that it effectively describes the dynamic behavior of a viscoelastic material in a wide frequency range. For this purpose, several dynamic shear tests were carried out at different temperatures and for different vibration frequencies, and then the values of all four parameters of the model under consideration were identified.

Several conclusions were drawn on the basis of the results of the experimental studies and the conducted analyses. First, the four-parameter, fractional Zener model can very well describe the dynamic behavior of a viscoelastic material at any temperature in the frequency range that can be achieved in a real laboratory test. However, for a wider frequency range, the description of the rheological properties of the tested material using the Zener model is less accurate.

The compression test showed that the tested material is isotropic. The isotropy was also confirmed in the shear test, although the results of these tests are not reported in the paper. Most of the experiments involved shear tests, because then the viscoelastic material has the greatest energy dissipation capacity and is used in vibration dampers in this configuration.

Based on the created Cole–Cole plot and the wicket plot, it could be assumed that the tested material is thermo-rheologically simple. This assumption made it possible to determine the shift factor function, which was then used to create the so-called master curves separately for the storage and the loss modules. Moreover, some parametrical analyzes demonstrated how individual parameters of the model influence the shape of the functions of the storage and loss modules.

It can be concluded that the four-parameter fractional Zener model well describes the function of the storage modulus for the selected viscoelastic material over the entire wide frequency range (i.e., twelve decades). On the other hand, the Zener model describes the loss modulus well only for the middle three decades of frequency, as there are discrepancies both for the lowest range and for the highest frequency range.

For the storage modulus, a comparable mean error value was obtained for the entire frequency range ($R_{a1}=8.18\%$) and for its middle range ($R_{r1}=10.75\%$). Whereas for the loss modulus, the fit in the entire frequency range is burdened with a large error ($R_{a2}=40.25\%$), and for its central part the error is much smaller ($R_{r2}=15.37\%$).

Moreover, the parametric analysis of the solutions showed that it is impossible to fit the loss modulus function in the entire frequency range (i.e., for twelve decades).

In general, decreasing or increasing the value of a model parameter causes a significant increase in errors in fitting the module functions. Only one case is different, increasing the $k_{o}$ parameter reduces the error for the $G_{2}$ module by less than a percent, but at the same time increases the error for the $G_{1}$ module by more than 6 percent. Therefore, in general, there is a deterioration in the adjustment of the module functions to the results of the experiment.

Probably, to obtain a better fit of the loss modulus function, one would need to apply a rheological model with more parameters describing it.

Future work will focus on applying the proposed approach to axial compression and tensile tests, testing other viscoelastic materials and extending the research to other, more complex rheological models.

## Figures and Tables

**Figure 1 materials-14-07024-f001:**
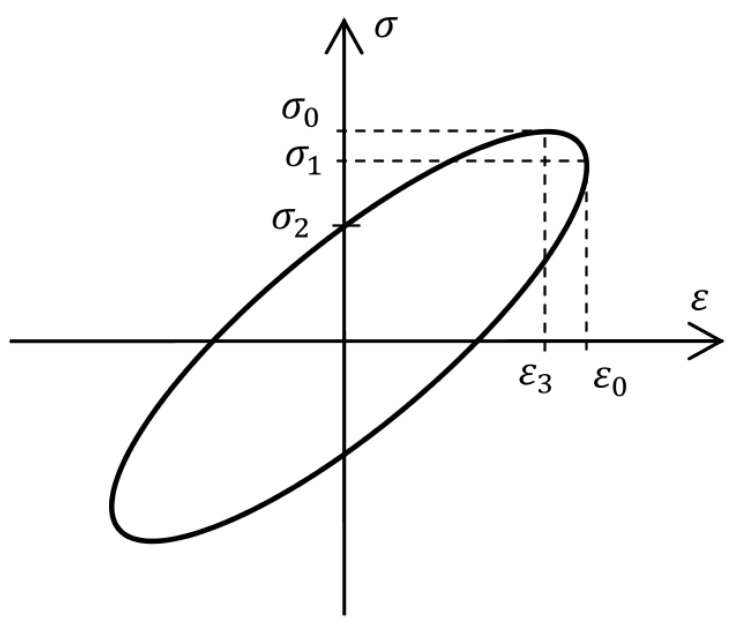
Example of hysteresis loop for VE material, stress–strain curve.

**Figure 2 materials-14-07024-f002:**
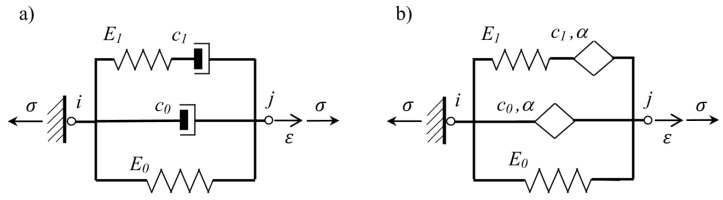
Complex models of VE material: (**a**) classical, and (**b**) fractional.

**Figure 3 materials-14-07024-f003:**
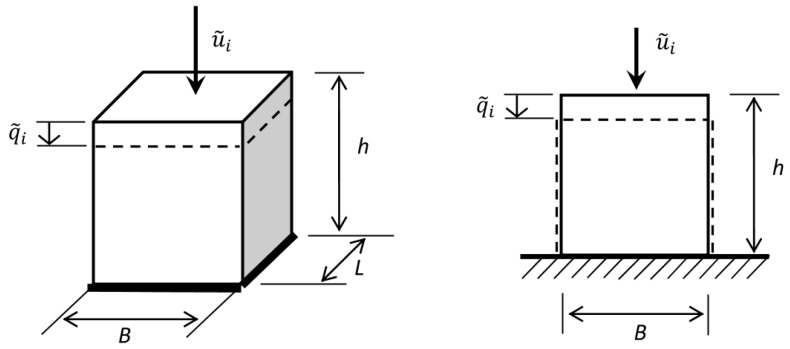
The desired dimensions of the sample tested in the compression test.

**Figure 4 materials-14-07024-f004:**
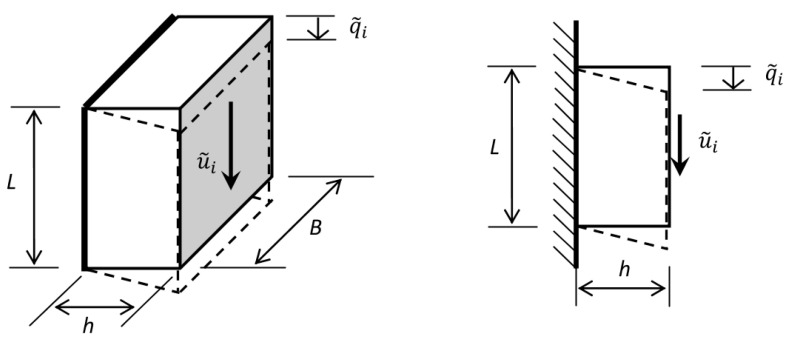
The desired dimensions of the sample tested in the shear test.

**Figure 5 materials-14-07024-f005:**
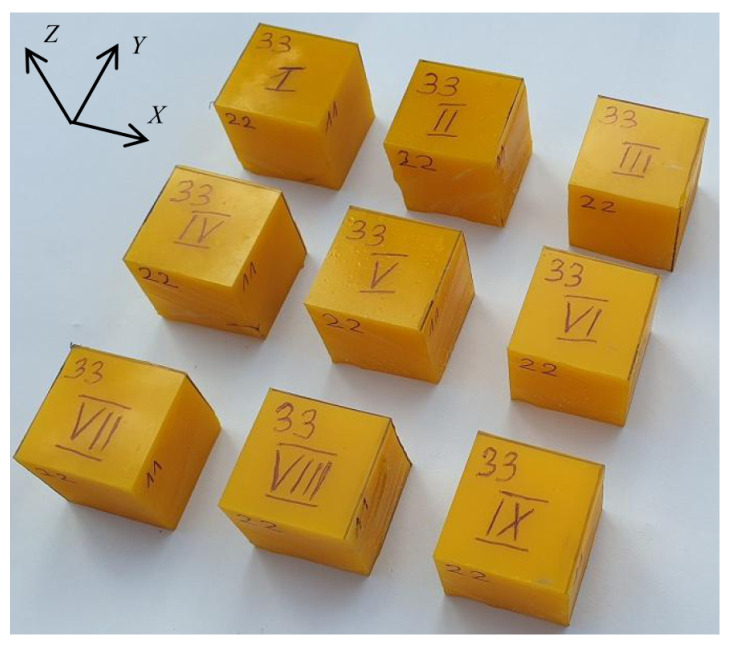
Viscoelastic material samples for compression test.

**Figure 6 materials-14-07024-f006:**
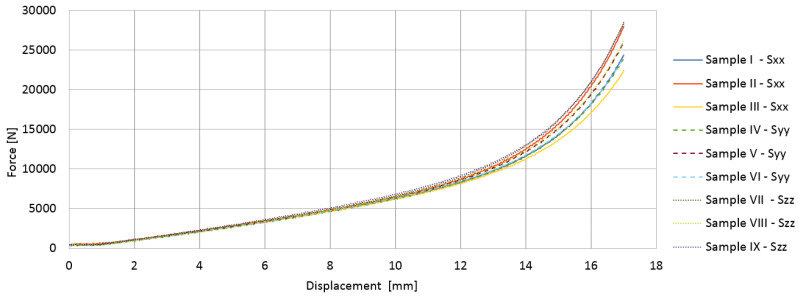
Force vs. displacement for compression tests of VE material in different directions.

**Figure 7 materials-14-07024-f007:**
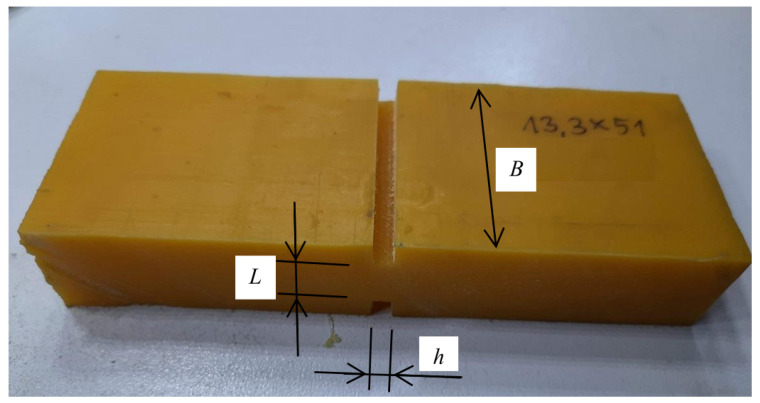
A sample of a viscoelastic material with a notch in the center.

**Figure 8 materials-14-07024-f008:**
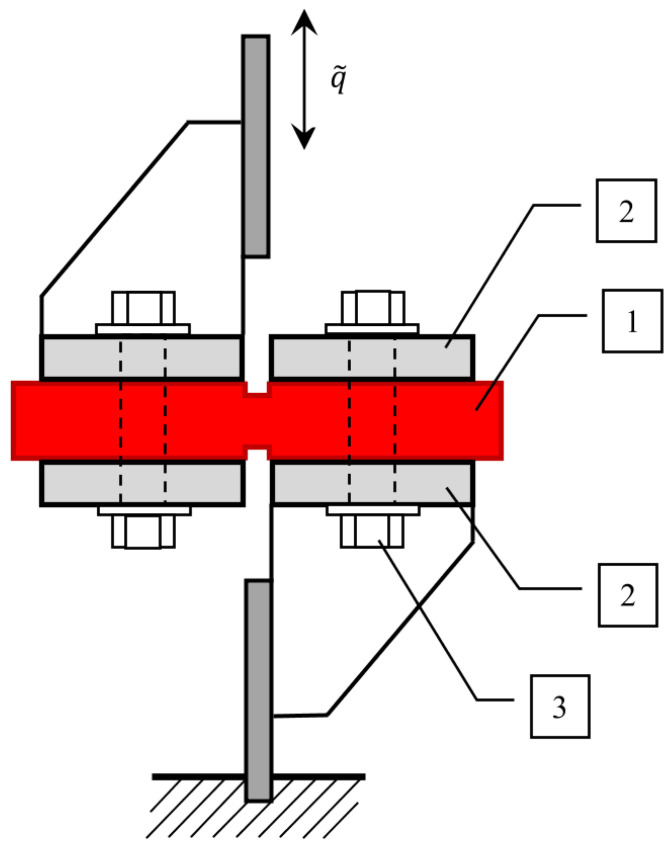
Special device to hold the sample in the transverse shear test. 1—VE material layer, 2—Steel sheets, 3—Clamping screw.

**Figure 9 materials-14-07024-f009:**
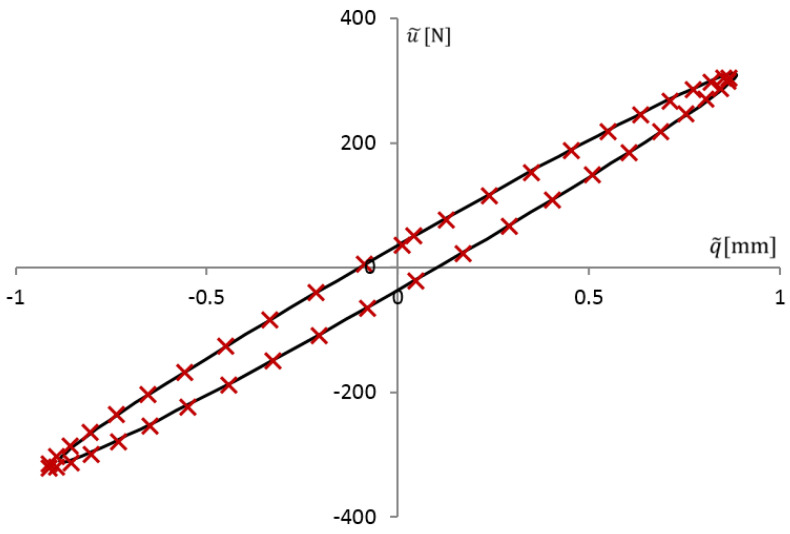
Relationship between force and displacement in the shear test: points-results of laboratory measurements, solid line-approximating function.

**Figure 10 materials-14-07024-f010:**
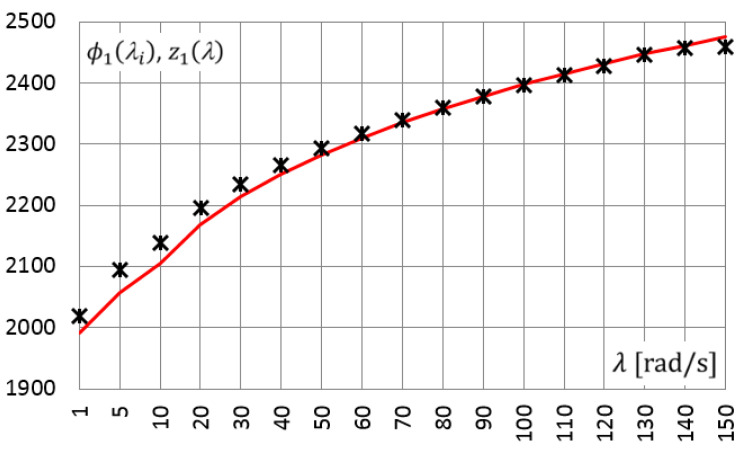
Functional relationships for selected frequencies: $\phi_{1}\left(\lambda_{i}\right)$—obtained from laboratory test (points), $z_{1}\left(\lambda \right)$—derived for Zener model (solid line).

**Figure 11 materials-14-07024-f011:**
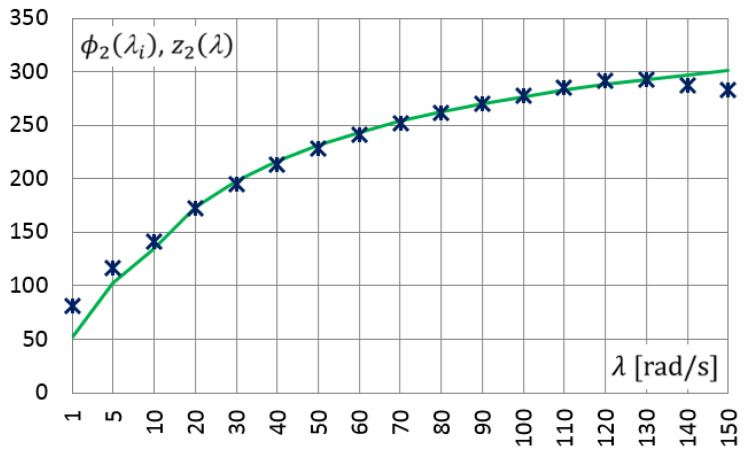
Functional relationships for selected frequencies: $\phi_{2}\left(\lambda_{i}\right)$—obtained from laboratory test (points), $z_{2}\left(\lambda \right)$—derived for Zener model (solid line).

**Figure 12 materials-14-07024-f012:**
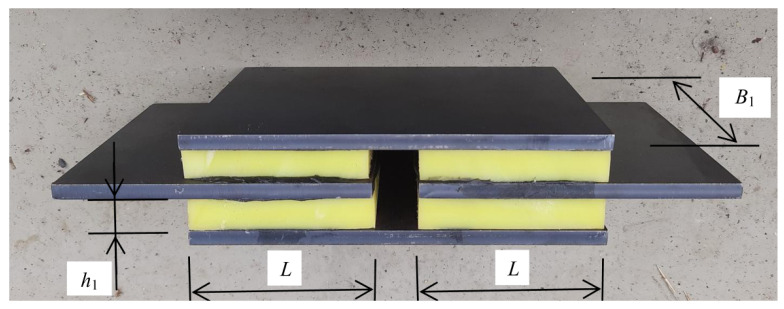
Layers of viscoelastic material glued to the steel plates.

**Figure 13 materials-14-07024-f013:**
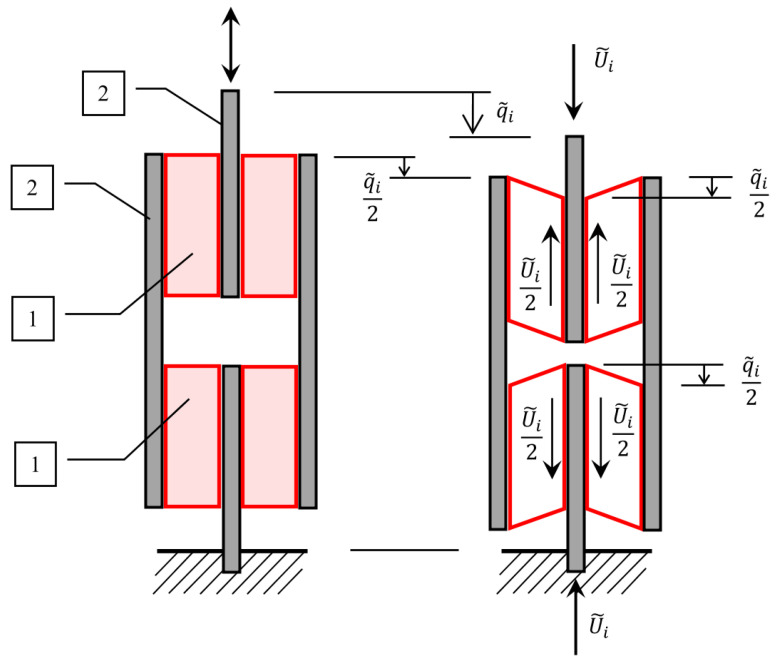
Distribution of forces and displacements in the longitudinal shear test. 1—VE material layer, 2—steel sheets.

**Figure 14 materials-14-07024-f014:**
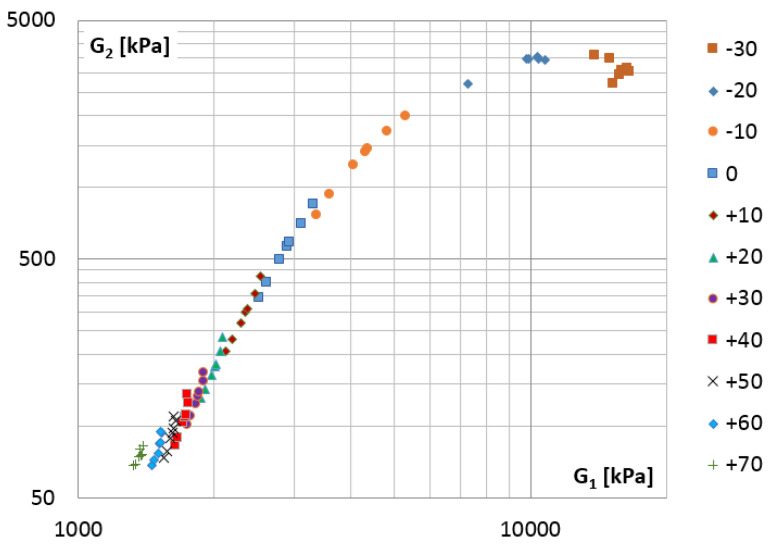
The Cole–Cole plot, the loss modulus $G_{2}\left(\lambda_{i}\right)$ vs. storage modulus $G_{1}\left(\lambda_{i}\right)$ for selected temperatures.

**Figure 15 materials-14-07024-f015:**
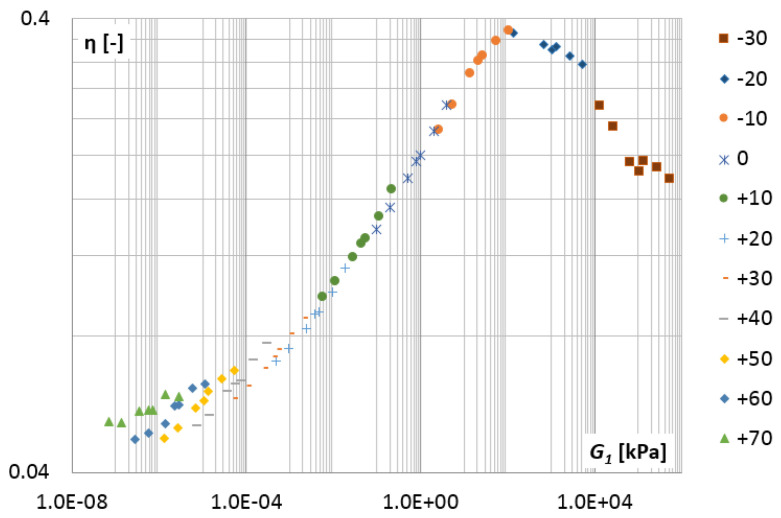
The wicket plot, the loss factor η=tan(δ) vs. storage modulus $G_{1}\left(\lambda_{i}\right)$ for selected temperatures.

**Figure 16 materials-14-07024-f016:**
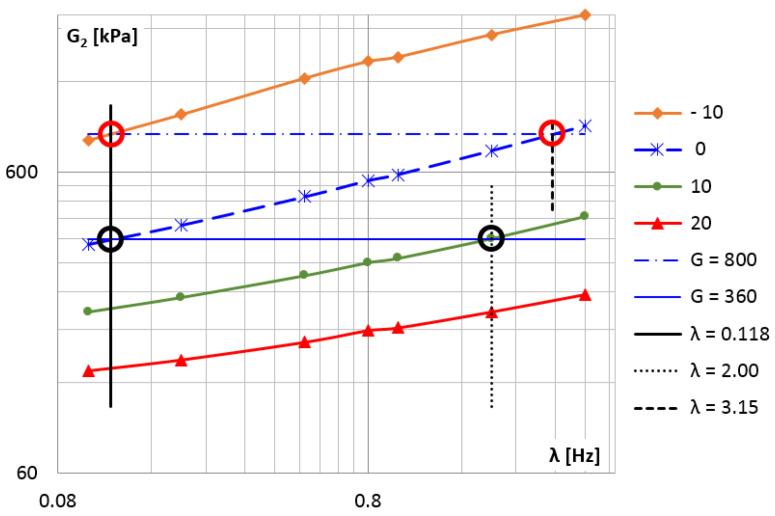
The loss modulus diagram for selected temperatures in the frequency domain.

**Figure 17 materials-14-07024-f017:**
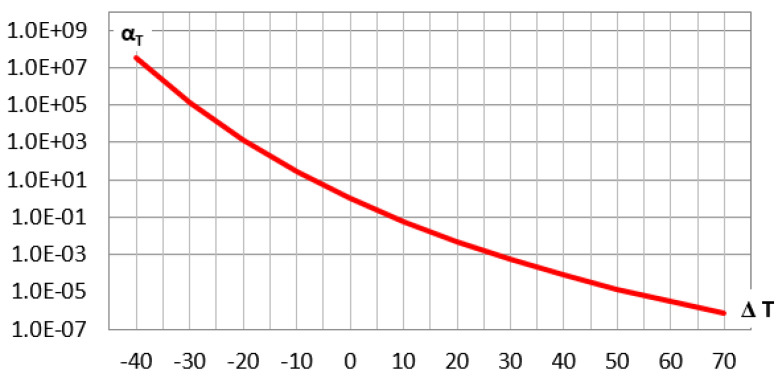
The shift factor $\alpha_{T}$ diagram as a function of temperature increment $▵T_{i}=T_{i}-T_{0}$.

**Figure 18 materials-14-07024-f018:**
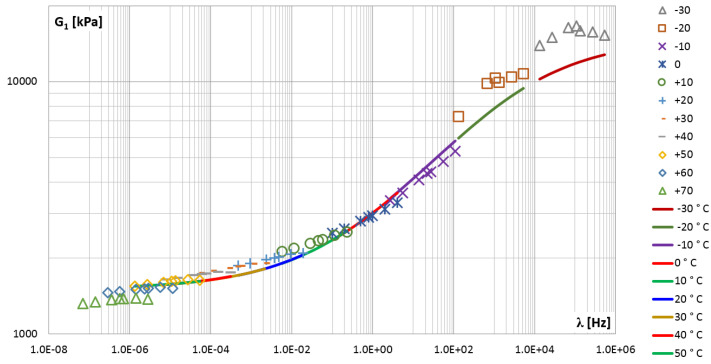
The shear storage modulus $G_{1}$ vs. frequency-master curve.

**Figure 19 materials-14-07024-f019:**
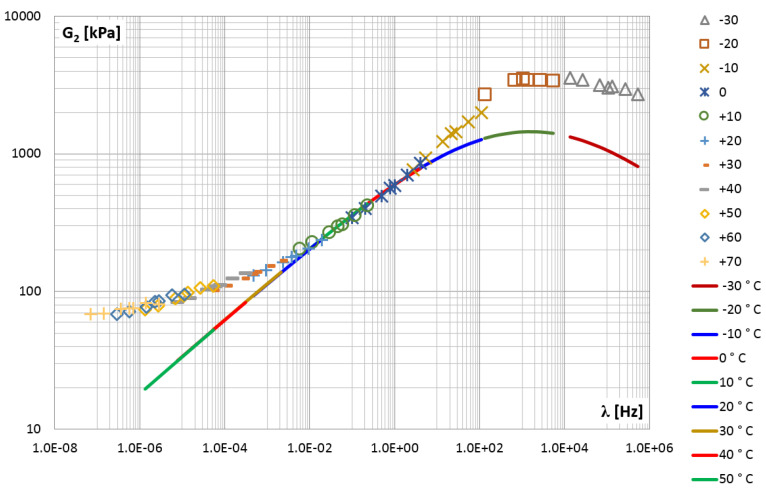
The shear loss modulus $G_{2}$ vs. frequency—master curve.

**Figure 20 materials-14-07024-f020:**
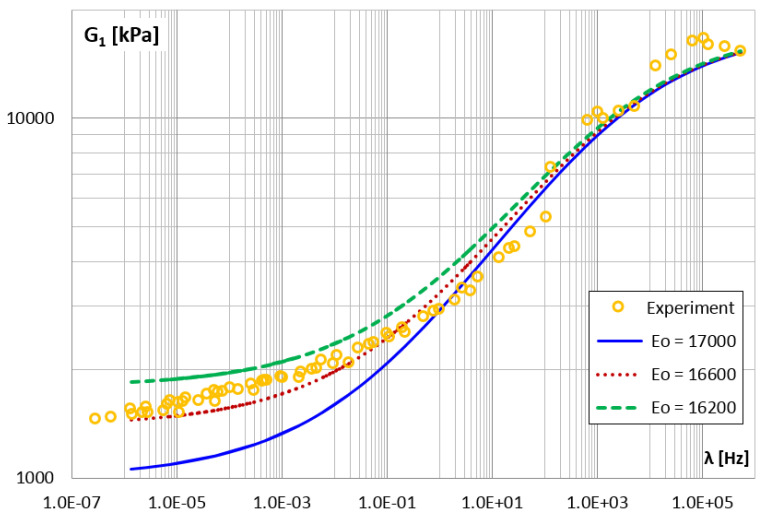
The shear storage modulus $G_{1}$ vs. frequency for different values of parameter $E_{0}$.

**Figure 21 materials-14-07024-f021:**
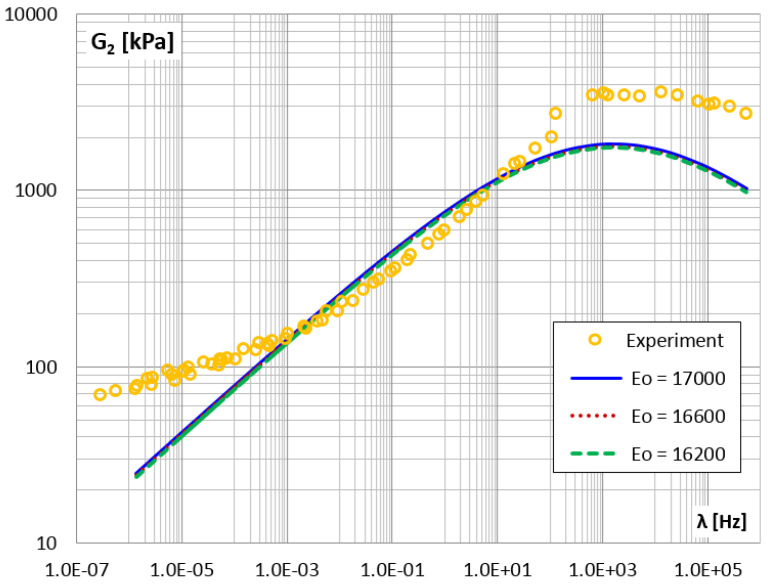
The shear loss modulus $G_{2}$ vs. frequency for different values of parameter $E_{0}$.

**Figure 22 materials-14-07024-f022:**
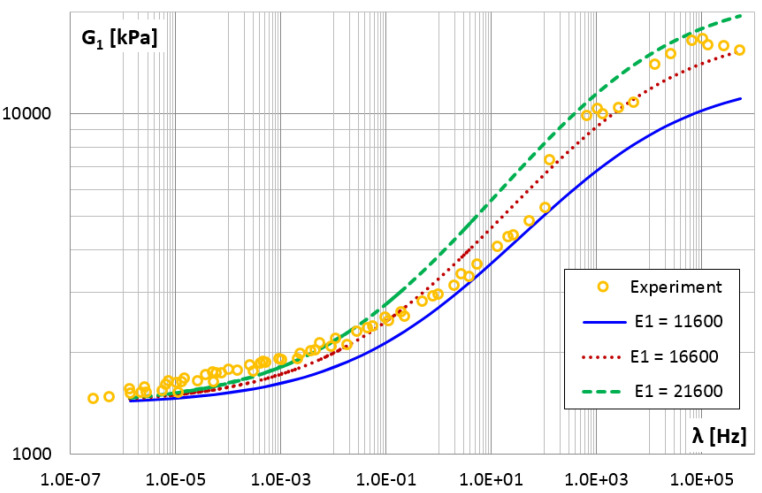
The shear storage modulus $G_{1}$ vs. frequency for different values of parameter $E_{1}$.

**Figure 23 materials-14-07024-f023:**
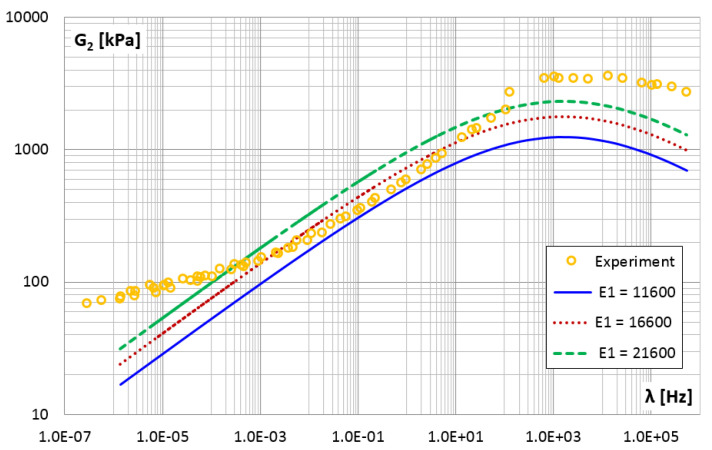
The shear loss modulus $G_{2}$ vs. frequency for different values of parameter $E_{1}$.

**Figure 24 materials-14-07024-f024:**
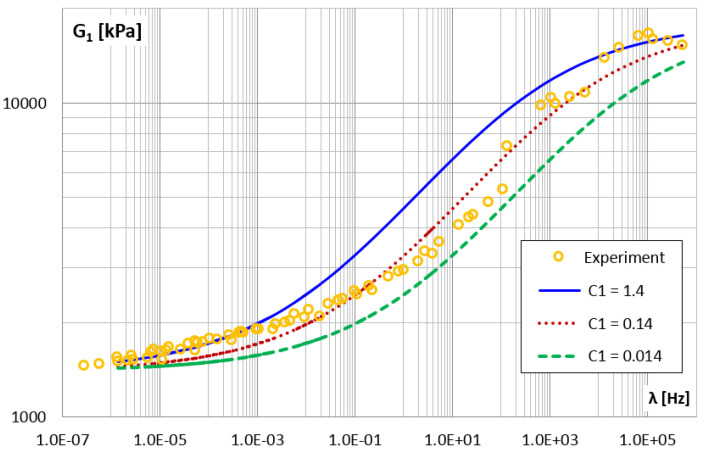
The shear storage modulus $G_{1}$ vs. frequency for different values of parameter $c_{1}$.

**Figure 25 materials-14-07024-f025:**
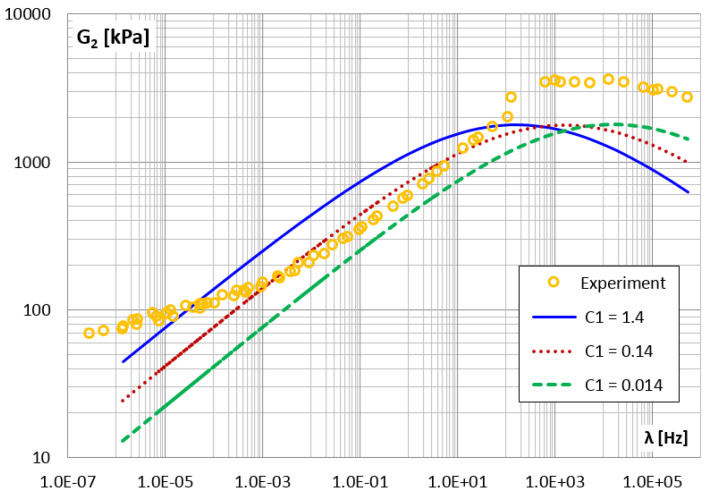
The shear loss modulus $G_{2}$ vs. frequency for different values of parameter $c_{1}$.

**Figure 26 materials-14-07024-f026:**
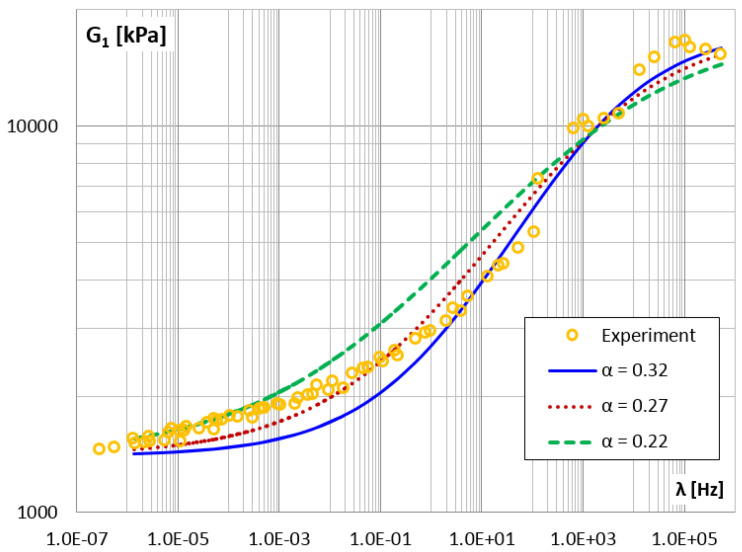
The shear storage modulus $G_{1}$ vs. frequency for different values of parameter α.

**Figure 27 materials-14-07024-f027:**
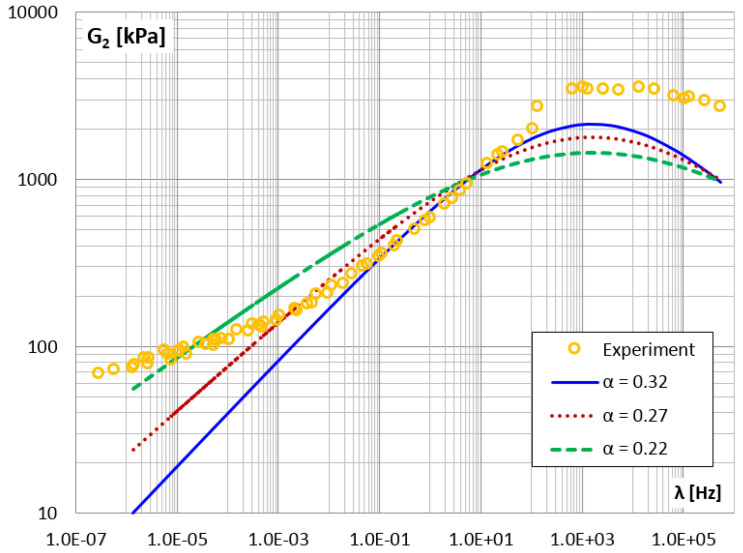
The shear loss modulus $G_{2}$ vs. frequency for different values of parameter α.

**Figure 28 materials-14-07024-f028:**
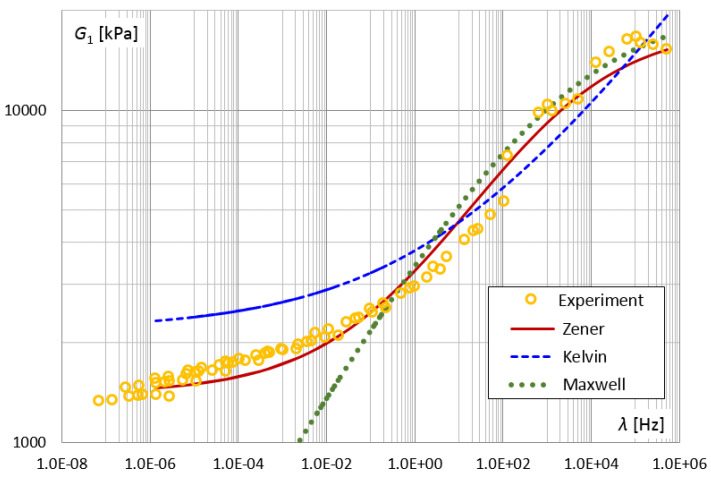
The shear storage modulus $G_{1}$ vs. frequency for different fractional models.

**Figure 29 materials-14-07024-f029:**
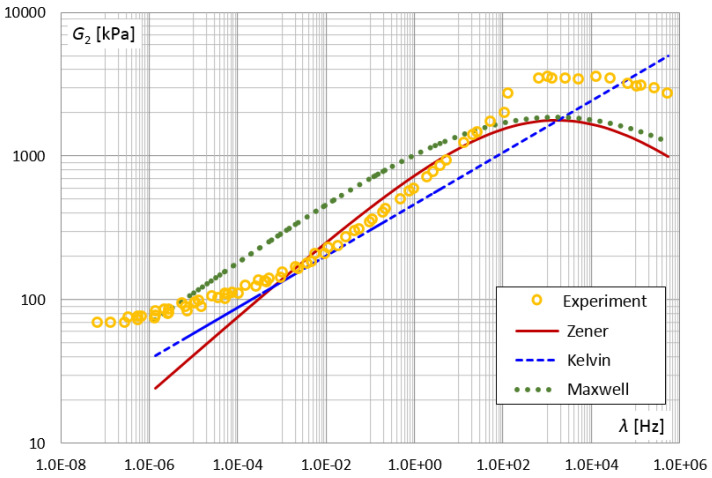
The shear loss modulus $G_{2}$ vs. frequency for different fractional models.

**Table 1 materials-14-07024-t001:** The results of the approximation of laboratory measurements (λ=140 rad/s).

Amplitudes of Approximating Functions		Functional Relationships	
${\tilde{q}}_{c}$ [mm]	0.00556	${\tilde{\phi }}_{1}\;\left[\frac{N}{m}\right]$	349,503
${\tilde{q}}_{s}$ [mm]	0.88536		
${\tilde{u}}_{c}\;\left[N\right]$	37.98	${\tilde{\phi }}_{2}\;\left[\frac{N}{m}\right]$	40,700
${\tilde{u}}_{s}\;\left[N\right]$	309.21		

**Table 2 materials-14-07024-t002:** Approximation results given for the stress-strain relationship (λ=140 rad/s).

Amplitudes of Approximating Functions		Functional Relationships	
${\tilde{\gamma }}_{c}\;\left[-\right]$	0.00116	$\phi_{1}$ [kPa]	2454.02
${\tilde{\gamma }}_{s}\;\left[-\right]$	0.18445		
${\tilde{\tau }}_{c}$ [Pa]	55,551	$\phi_{2}$ [kPa]	285.78
${\tilde{\tau }}_{s}$ [Pa]	452,313		

**Table 3 materials-14-07024-t003:** Model parameter values selected for additional calculations.

	Model Parameter		Mean Relative Error	
			**$R_{r1}$[%]**	**$R_{r2}$[%]**
$E_{0}$	$\left[\frac{kN}{m}\right]$	1000	18.59	16.23
		**1400**	**10.75**	**15.37**
		1800	16.97	14.52
$E_{1}$	$\left[\frac{kN}{m}\right]$	11,600	17.45	37.99
		**16,600**	**10.75**	**15.37**
		21,600	32.38	73.82
$c_{1}$	$\left[\frac{s\cdot kN}{m}\right]$	0.014	11.65	25.64
		**0.14**	**10.75**	**15.37**
		1.40	18.52	38.46
α	[−]	0.22	22.36	40.97
		**0.27**	**10.75**	**15.37**
		0.32	13.25	22.39

## Data Availability

Data sharing not available.

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
