# Peer review of "Identification of the Fractional Zener Model Parameters for a Viscoelastic Material over a Wide Range of Frequencies and Temperatures"

_materials, 2021, doi:10.3390/ma14227024_

Round 1

Reviewer 1 Report

The submitted article is original and contain interesting discussions. I recommend it for publication after minor correction. 

  1. I think the introduction section contain many old studies and can be replaced by new ones: https://doi.org/10.5459/bnzsee.48.2.100-117; https://ascelibrary.org/doi/abs/10.1061/%28ASCE%29SC.1943-5576.0000563; Mirrashid, M. and Naderpour, H., 2021. Innovative Computational Intelligence-Based Model for Vulnerability Assessment of RC Frames Subject to Seismic Sequence. Journal of Structural Engineering147(3), p.04020350.Naderpour, H., Mirrashid, M. and Parsa, P., 2021. Failure mode prediction of reinforced concrete columns using machine learning methods. Engineering Structures248, p.113263.Mirrashid, M. and Naderpour, H., 2021. Recent trends in prediction of concrete elements behavior using soft computing (2010–2020). Archives of Computational Methods in Engineering, Vol. 28, pp.3307-3327.
  2. Figures can be drawn by MATLAB or OriginPro to have a better quality. 
  3. Conclusions must specific and based on the results, please avoid adding generic ones. 

Reviewer 2 Report

The comparison of the experimental data and the curves obtained from formulas is rather qualitative. Some parts fit quite well, others are way off. From my point of view a more quantitative analysis of the parameters is needed. Are the "selected" parameter values (p.21) really the best values obtained from an optimization process or are they just the best values which were used for some of the curves? What is the error which is left for these optimal values? Are there additional interpretations/reasons for the deviations of the results (esp. in the very high frequency range)?

Author Response

Dear Reviewer,

The authors would like to thank the reviewer for his comments that help improve the manuscript. The new version of the manuscript includes all of the reviewer's comments. Based on these comments, the paper has been integrated and upgraded. All changes are written in red.

In detail, these comments are reported point by point below.

1. The comparison of the experimental data and the curves obtained from formulas is rather qualitative. Some parts fit quite well, others are way off. From my point of view a more quantitative analysis of the parameters is needed.

Quantitative analyzes concerning the error of fitting the function describing the model to the empirical data have been added.

2. Are the "selected" parameter values (p.21) really the best values obtained from an optimization process or are they just the best values which were used for some of the curves? What is the error which is left for these optimal values?

In the PSO method, after repeating the calculations several times, the parameters that are considered to be the best matching solution are selected, but there is no certainty that the obtained solution is a global and not just a local minimum. Quantitative analyzes concerning the error of fitting have been added.

3. Are there additional interpretations/reasons for the deviations of the results (esp. in the very high frequency range)?

According to the authors, the only reason for the deviation of the results is too few parameters in the model for this particular viscoelastic material. This thesis is to be confirmed by further studies using more complex models.

Best regards

Zdzislaw Pawlak

Arkadiusz Denisiewicz

Reviewer 3 Report

Viscoelastic materials are widely applied in damping systems due to the good rheological properties. Here, the mechanical model is very important to describe the dynamic behavior. Authors developed a four-parameter, fractional Zener model and its parameters of the viscoelastic material have been studied. Some interesting works have been carried out. However, the current form of this study cannot be acceptable. Some aspects as listed below:

  1. What are the advantages of Zener model for describing the dynamic behavior? It is suggested to compare the results with other related models.
  2. More details about the tested viscoelastic material samples should be given.
  3. What is the scope of the model for different viscoelastic materials.

Author Response

Dear Reviewer,

The authors would like to thank the reviewer for his comments that help improve the manuscript. The new version of the manuscript includes all of the reviewer's comments. Based on these comments, the paper has been integrated and upgraded. All changes are written in red.

In detail, these comments are reported point by point below.

1. What are the advantages of Zener model for describing the dynamic behavior? It is suggested to compare the results with other related models.

The results of additional analyzes concerning the three-parameter Kelvin and Maxwell models were added.

2. More details about the tested viscoelastic material samples should be given.

Additional information on the viscoelastic material has been added.

3. What is the scope of the model for different viscoelastic materials.

Each viscoelastic material has different rheological properties that the selected model may describe better or worse. Therefore, when examining a specific material, it is not only necessary to identify the parameters of the model, but also to select the appropriate model.

Best regards

Zdzislaw Pawlak

Arkadiusz Denisiewicz

Round 2

Reviewer 2 Report

The revised version is fine.